# Associations between patterns in comorbid diagnostic trajectories of individuals with schizophrenia and etiological factors

Morten Dybdahl Krebs[1,2], Gonçalo Espregueira Themudo [1,2], Michael Eriksen Benros [3,4], Ole Mors[2,5], Anders D. Børglum [2,6,7], David Hougaard [2,8], Preben Bo Mortensen[2,7,9], Merete Nordentoft[2,3], Michael J. Gandal [10,11,12], Chun Chieh Fan [1,2,13], Daniel H. Geschwind [10,11,12,14], Andrew J. Schork [1,2,15], Thomas Werge [1,2,16,17,19]✉ & Wesley K. Thompson[1,2,18,19]

Schizophrenia is a heterogeneous disorder, exhibiting variability in presentation and outcomes that complicate treatment and recovery. To explore this heterogeneity, we leverage the comprehensive Danish health registries to conduct a prospective, longitudinal study from birth of 5432 individuals who would ultimately be diagnosed with schizophrenia, building individual trajectories that represent sequences of comorbid diagnoses, and describing patterns in the individual-level variability. We show that psychiatric comorbidity is prevalent among individuals with schizophrenia (82%) and multi-morbidity occur more frequently in specific, time-ordered pairs. Three latent factors capture 79% of variation in longitudinal comorbidity and broadly relate to the number of co-occurring diagnoses, the presence of child versus adult comorbidities and substance abuse. Clustering of the factor scores revealed five stable clusters of individuals, associated with specific risk factors and outcomes. The presentation and course of schizophrenia may be associated with heterogeneity in etiological factors including family history of mental disorders.

[1] Institute of Biological Psychiatry, Mental Health Centre Sct Hans, Mental Health Services Capital Region of Denmark, Copenhagen, Denmark. [2] iPSYCH, The Lundbeck Foundation Initiative for Integrative Psychiatric Research, Aarhus, Denmark. [3] Copenhagen Research Centre for Mental Health, Mental Health Centre Copenhagen, Copenhagen University Hospital, Copenhagen, Denmark. [4] Department of Immunology and Microbiology, Faculty of Health and Medical Sciences, University of Copenhagen, Copenhagen, Denmark. [5] Aarhus University Hospital, Risskov, Denmark. [6] Department of Biomedicine and iSEQ-Centre for Integrative Sequencing, Aarhus University, Aarhus, Denmark. [7] Center for Genomics and Personalized Medicine, Aarhus University, Aarhus, Denmark. [8] Department for Congenital Disorders, Statens Serum Institut, Copenhagen, Denmark. [9] National Centre for Register-Based Research, Aarhus University, Business and Social Sciences, Aarhus, Denmark. [10] Department of Neurology, University of California, Los Angeles, Los Angeles, CA, USA. [11] Department of Human Genetics, University of California, Los Angeles, Los Angeles, CA, USA. [12] Center for Autism Research and Treatment, Semel Institute, David Geffen School of Medicine, University of California, Los Angeles, Los Angeles, CA, USA. [13] Center for Human Development, University of California, San Diego, CA, USA. [14] Program in Neurobehavioral Genetics, Semel Institute, David Geffen School of Medicine, University of California Los Angeles, Los Angeles, CA, USA. [15] Neurogenomics Division, The Translational Genomics Research Institute (TGEN), Phoenix, AZ, USA. [16] Center for GeoGenetics, Globe Institute, Faculty of Health and Medical Sciences, University of Copenhagen, Copenhagen, Denmark. [17] Department of Clinical Medicine, Faculty of Health and Medical Sciences, University of Copenhagen, Copenhagen, Denmark. [18] Division of Biostatistics and Department of Radiology, Population Neuroscience and Genetics Lab, University of California, San Diego, CA 92093, USA. [19] These authors contributed equally: Thomas Werge, Wesley K. Thompson. ✉email: thomas.werge@regionh.dk

Psychiatric disorders have been classified for close to a century using a categorical and syndromic approach based on subjectively observed and reported psychiatric symptoms, rather than objective biomarkers, which limits their specificity and utility for guiding interventions[1,2]. Schizophrenia, in particular, can be seen as the canonical example of a syndrome with varying clinical presentations, inconsistent treatment response, and longitudinal prognostic instability[3–5]. From one perspective, the heterogeneity of schizophrenia implies that the diagnosis is too broad because it is sensitive to various unique clinical presentations. From another perspective, however, it is too narrow, as the diagnosis is often not unitary, but regularly appears in constellation with multiple comorbid symptoms, diagnoses, or pathologies[1,6–10]. This necessitates questions into the nature of this phenomenon—does heterogeneity in schizophrenia merely reflect a constellation of random phenomena with distinct etiologies?

The idea of subtypes within schizophrenia dates back more than a century[3], but previous subtypes were used infrequently[11], had modest longitudinal stability[12], inadequately described symptom heterogeneity, and have been abandoned in DSM-5[11] and ICD-11[13]. This reflects initiatives to describe the clinical heterogeneity in schizophrenia on symptom dimensions[14] and factor analysis of symptom dimensions[15]. However, these approaches have been limited by their cross-sectional design (e.g., Picardi et al.[15]), short follow-up (e.g., Dwyer et al.[16]), or retrospective data collection (e.g., Strous et al.[17]). Despite this, interest in describing heterogeneity in schizophrenia persists[7] and longitudinal cohorts, with broad phenotyping, and representative ascertainment are well poised to make important contributions to this topic.

Although there is substantial pleiotropy, or sharing of genetic risk factors, across different psychiatric disorders[18] genetic differences are also present[19,20], which indicates that different pathological mechanisms are implicated, at least to some extent, depending on the psychiatric diagnoses[7,21]. Since psychiatric comorbidity is very common in schizophrenia[8] different comorbid diagnoses might reflect differences in underlying biology and therefore there is a need for broader investigations into the nature of clinical heterogeneity in schizophrenia, especially with respect to longitudinal outcomes and comorbid diagnostic patterns.

The Danish health registers offer a unique perspective on life-course heterogeneity in the clinical presentations of individuals with schizophrenia. Established in 1968, the Danish registration system[22–24] has provided nearly complete coverage of the health service usage of the complete population of Denmark for more than 50 years, including psychiatric hospital contacts[23]. The Psychiatric Central Research Register (PCRR)[23] follows the population from birth in a longitudinal and prospective manner, providing a unique, time-stamped, and reliable[23,25–28] diagnoses for an individual at each hospital contact. These data, then, more closely reflect real-world clinical practice than retrospective case-control diagnoses because they objectively catalog preceding and succeeding psychiatric contacts. Previous studies have used these powerful data to define rates of comorbid diagnoses in psychiatry[8] and describe specific patterns of transitions among diagnoses to demonstrate difficulties classifying individuals with mental disorders at first admission[29]. Danish register data have also been used to describe both prodromal states[30] and pre-morbid traits in individuals later diagnosed with schizophrenia[30,31], and to record longitudinal stability in patterns of comorbidity[32]. A systematic, data-driven study of life-course patterns of comorbid diagnoses and their relation to etiological factors has not been pursued but could contribute greatly to how we understand heterogeneity within schizophrenia.

We hypothesized that predictable trends exist in the patterns of longitudinal, comorbid psychiatric diagnoses among individuals with schizophrenia. To test this hypothesis, we obtained complete psychiatric hospitalization records for all persons born in Denmark between 1981 and 2002 that had received a diagnosis of schizophrenia by the end of 2012 ($N = 5432$). We quantified inter-individual differences in comorbidity trajectories using Sequence Analysis[33] and examined the structure in the differences using multidimensional scaling (MDS) and cluster analysis of the resulting first principal MDS dimensions. To investigate whether the heterogeneity within clinically defined schizophrenia (ICD-10 F20.0-F20.9) is linked to etiological heterogeneity, the individual patient projections onto the leading principal dimensions of trajectory dissimilarity from the MDS analysis were tested for association with known risk factors and clinical outcomes. This could help uncover biological heterogeneity and motivate more personalized clinical care to improve outcomes in schizophrenia.

## Results

From the Danish patient registers, all individuals born between 1981 and 2002 and diagnosed with schizophrenia (ICD10: F20.0-20.9) by the end of 2012 ($N = 5432$) were identified together with 10,864 random age- and sex-matched population controls and followed until December 31st, 2016 (Table 1). Comorbid psychiatric diagnoses made prior, simultaneous, or subsequent to the first schizophrenia diagnosis were observed for 4456 of the 5432 participants (82%), with substance abuse, mood disorders, and personality disorders being the most prevalent diagnoses. Seventy percent ($N = 3790$) of all participants received at least one other psychiatric diagnosis before their first schizophrenia diagnosis (Table 1). The prevalence of other psychiatric diagnoses in individuals diagnosed with schizophrenia was more than 5-fold higher at age of censoring than among the population controls (relative risk (RR) ranging from 6.2 (95% confidence interval (CI): 5.0–7.6) for eating disorders to 18.7 (CI: 16.4–21.3) for substance abuse; Table 1). The cumulative incidence of receiving specific comorbid psychiatric diagnoses[34] (Table 1 and Supplementary Table S1) is shown in Fig. 1, both when the diagnosis occurs before and after the initial schizophrenia diagnosis, and presented along with the corresponding incidence in controls. A substantial proportion of childhood disorders, defined by a typically early onset in population cohorts, were diagnosed after the first diagnosis of schizophrenia, such that they occurred in adolescence or adulthood.

Hazard ratios (HRs) for 22 of 56 pairs of comorbid diagnoses (not including schizophrenia) for individuals with schizophrenia were significant ($P < 0.00089$; Fig. 2 and Supplementary Table S2) suggesting temporal structure in the ordering of comorbidities. Both increased and decreased hazards were observed within the schizophrenia patient cohort. As an example, a comorbid diagnosis of mood disorders increased the hazard of subsequent diagnoses for personality disorders, eating disorders, and anxiety and obsessive-compulsive disorders, while a diagnosis of autism spectrum disorder reduced the hazard of a subsequent diagnosis of substance abuse. For some pairs of disorders, each increased the later probability of a diagnosis of the other (e.g., substance use disorder diagnoses increases the hazard of personality disorder diagnoses and vice versa), whereas for other pairs the increase was unidirectional (e.g., mood disorder diagnoses increase the hazard of autism disorder diagnoses, but the reverse was not observed). This is in contrast to Plana-Ripoll et al.[8], who showed that in the general population all diagnoses increase the risk of all other diagnoses. We observe HRs significantly lower than 1 for many pairs, which indicates more specificity. Overall, these results demonstrate that comorbidities

| Table 1 Characteristics of the population-wide cohort of individuals with schizophrenia and the age and sex-matched controls. | | | | | |
|---|---|---|---|---|---|
| | | | Case 2012 (N = 5432) | Matched (N = 10864) | |
| Sex | | | | | |
| Female | | | 2395 | 5127 | |
| Male | | | 3037 (56%) | 5737 (53%) | |
| Age | | | | | |
| 10–20 | | | 61 (1.1%) | 135 (1.2%) | |
| 20–25 | | | 781 (14%) | 1560 (14%) | |
| 25–30 | | | 2062 (38%) | 4043 (37%) | |
| 30–36 | | | 2528 (47%) | 5126 (47%) | |
| | | | | | Relative risk |
| Comorbid psychiatric diagnoses[a] | | | 4456 (82%) | 1277 (12%) | 7.0 (6.6–7.4) |
| F10-F19 | F10-F19 | Mental and behavioral disorders due to psychoactive substance use[b] | 2121 (39%) | 227 (2.1%) | 18.7 (16.4–21.3) |
| F40-F43.1 | F40-F43.1 | Neurotic disorders | 1166 (22%) | 368 (3.4%) | 6.3 (5.7–7.1) |
| F50 | F50 | Eating disorders | 370 (6.8%) | 120 (1.1%) | 6.2 (5.0–7.6) |
| F60 | F60 | Specific personality disorders | 1609 (30%) | 295 (2.7%) | 10.9 (9.7–12.3) |
| F70-F79 | F70-F79 | Mental retardation | 333 (6.1%) | 57 (0.5%) | 11.7 (8.8–15.4) |
| F84 | F84 | Pervasive developmental disorders | 389 (7.2%) | 101 (1.0%) | 7.7 (6.2–9.6) |
| F90-F98 | F90-F98 | Behavioral and emotional disorders with onset usually occurring in childhood and adolescence[c] | 1138 (21%) | 379 (3.5%) | 6.0 (5.4–6.7) |
| First non-F20 diagnosis before schizophrenia diagnosis | | | 3790 (70%) | – | – |

[a]For details on the translation from ICD-8 to ICD-10, see Supplementary Table S1.
[b]Substance abuse included 751 cases with alcohol-related, 1392 cases with cannabis-related, and 343 cases with abuse of other substances only.
[c]Behavioral and emotional disorders with onset usually occurring in childhood and adolescence included 598 cases with hyperkinetic disorders.

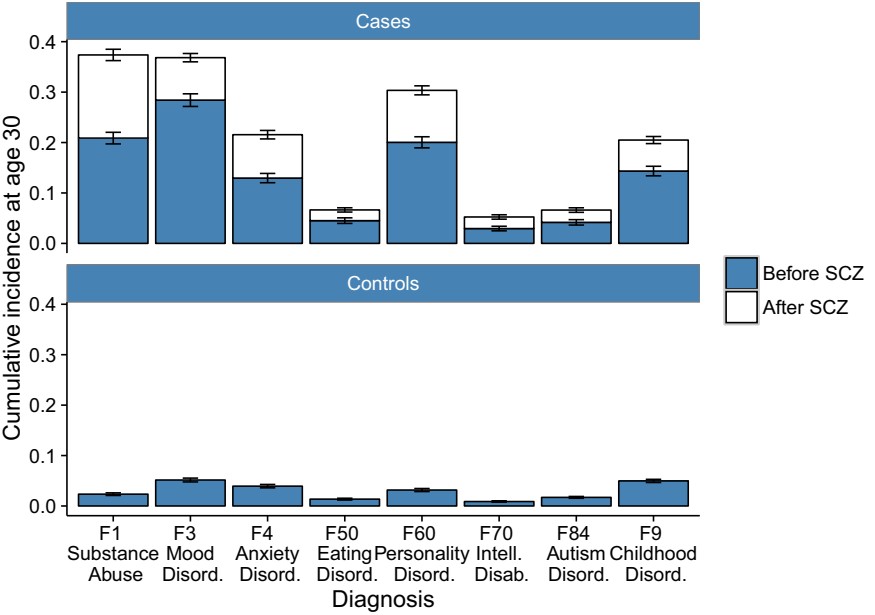

**Fig. 1 Cumulative incidence at age 30 of eight comorbid psychiatric disorder categories in a Danish cohort of 5432 individuals with schizophrenia and 10,864 random population controls.** The figure shows the cumulative incidence (CI) at age 30 of each of eight categories of comorbidities in individuals with schizophrenia and in the random population sample without schizophrenia. In blue are the CI before schizophrenia, in white are CI after schizophrenia. Error bars are 95% confidence intervals. The exact number of observed comorbidities are provided in Table 1.

within schizophrenia occur in specific time-dependent patterns and may reflect deeper, multiple-outcome structures that span the full follow-up period.

We constructed comorbid psychiatric trajectories (Fig. 3, and "Methods" section) for the 5432 individuals that summarized their life experience of diagnoses for multiple outcomes as "states" in one-year intervals from birth up to age 36 and used sequence analysis (SA)[33] (see "Methods" section) to define trajectory dissimilarity between all pairs of trajectories. Multidimensional scaling (MDS)[35] identified three principal components that explained 79%

of the variance in the dissimilarity matrix defined by SA (Supplementary Fig. S2). To test whether lowering time increment size would affect the results, we computed the sequence dissimilarities using 6 and 4 months increments and computed the correlation of the lower triangular entries of the resulting dissimilarity matrices with those obtained using 1-year increments. We found the correlations to be high ($r_{Pearson}$=0.9991, $r_{Pearson}$ = 0.9987, respectively) and therefore proceeded with the 1-year increment size.

To assess individual dimension loadings, we visualized the patterns of comorbidity at different quantiles of each dimension

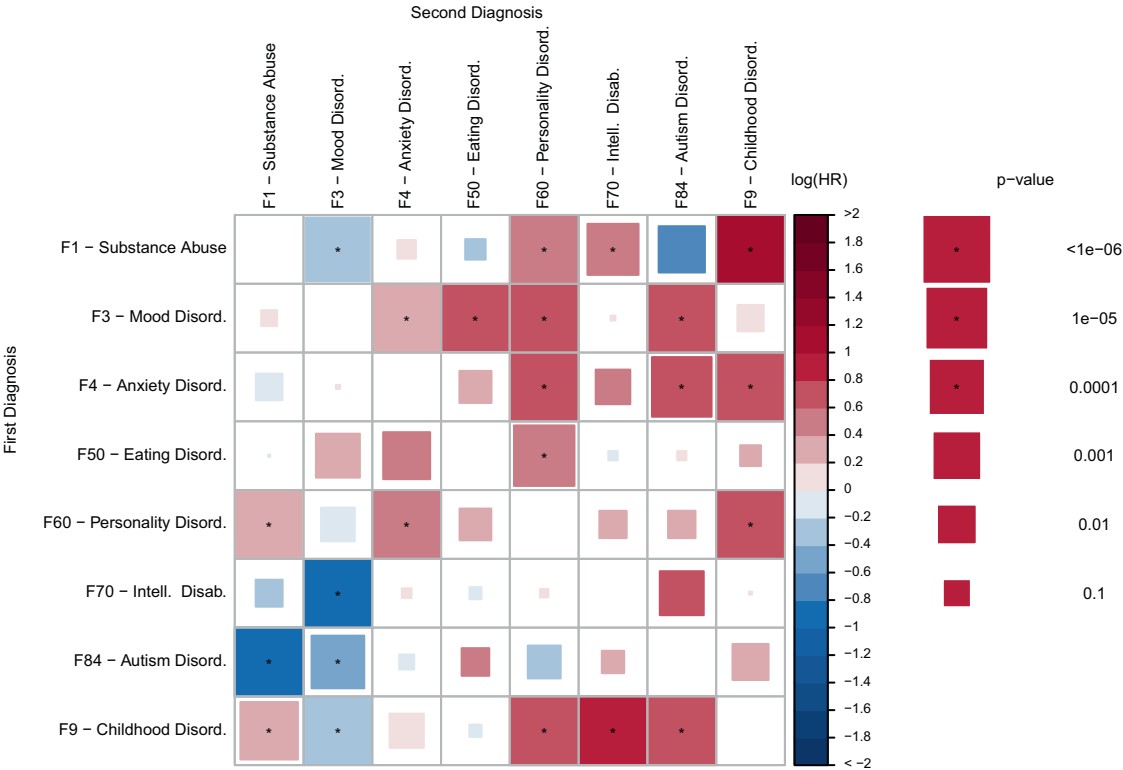

**Fig. 2 Hazard ratio of eight comorbid psychiatric disorder categories conditioned on a prior diagnosis of each of the seven other psychiatric disorder categories estimated in 5432 individuals with schizophrenia.** The figure displays the results of a Cox Proportional Hazard model. For each of the eight categories of psychiatric disorders (2nd diagnosis), the color indicates how the hazard depends on a previous diagnosis of each the seven other categories (1st diagnosis). All models are adjusted for age and sex. Square size indicates the uncorrected *P*-value of a two-sided Wald test with one degree of freedom. * indicates significance after a Bonferroni correction for 56 tests. *P*-values are unadjusted. The *N* (2 × 2 table) for each Cox regression is provided in Supplementary Table S2 along with the exact *P*-values.

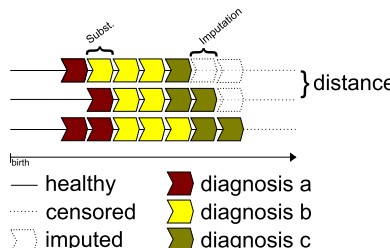

**Fig. 3 Schematic of the sequence analysis procedure used to compute the dissimilarity of psychiatric comorbidity trajectories.** Sequence dissimilarities were computed by alignment of the temporal sequence of diagnoses. Alignment could be obtained by substitutions, insertions, and deletions, which were each assigned a specific cost. Censored sequences were imputed based on previous diagnoses and assigned a cost of the probability-weighted average of possible states. Supplementary Fig. S1 illustrates the most common trajectories among individuals with schizophrenia.

(Supplementary Fig. S4) and, based on these patterns, we performed post-hoc linear and logistic regression analyses, predicting, in series, the constituent diagnoses (logistic regression), their total count (linear regression), and age at first diagnosis (linear regression), from each of the three principal MDS dimensions, independently. This showed that the first dimension captured the total number of comorbid disorders (mean of 0.76 additional diagnoses per standard deviation (sd) increase), the second-dimension distinguished adult from childhood comorbid disorders (odds ratio (OR) per sd increase, mood disorders: 0.034;

childhood disorders: 4.2), while the third-dimension characterized trajectories by the presence or absence of comorbid substance abuse, specifically (log(OR) per sd increase: 8.6) (Supplementary Figs. S4–S6). A jackknife stability test[36] found a stability coefficient of 0.9986 indicating the three principal MDS dimensions were highly stable. The three principal MDS dimensions, which summarize a majority of the variability in longitudinal comorbidity for schizophrenia, thereby had intuitive, interpretable, and clinically relevant factor loadings.

A conceptually similar and for some more intuitive presentation of latent, multivariate dimensions capturing heterogeneity is to define concrete groups that reflect a large portion of the variability in these abstract dimensions. We therefore subsequently used the principal MDS dimensions and Ward's agglomerative hierarchical clustering[37] to cluster individuals into subgroups. We found that nearly half of the variance in the three MDS scores was captured by five clusters ($R^2 = 0.48$; Supplementary Fig. S7) with distinct and clinically interpretable characteristics. Figure 4B shows the diagnosis that the plurality of individuals in each cluster had at each year of follow-up. Among the five clusters (1–5), cluster 1 ($N = 597$) contained individuals with comorbid childhood disorders (cumulative incidence (CI)$_{F7-F9}$: 100%), cluster 2 ($N = 1580$) contained individuals with multiple adult comorbidities (CI$_{F1-F6}$: 99%), cluster 3 ($N = 734$) contained individuals with only mood disorder comorbidities (CI$_{F3}$: 100%), cluster 4 ($N = 729$) contained individuals with comorbid substance abuse (CI$_{F1}$: 100%), and cluster 5 ($N = 1792$) contained individuals with little comorbidity (CI$_{F1-F9}$: 46%) (Fig. 4C). With the exception of cluster 3 (mood disorders-only) which did not separate clearly from cluster 5 (no-comorbidity)

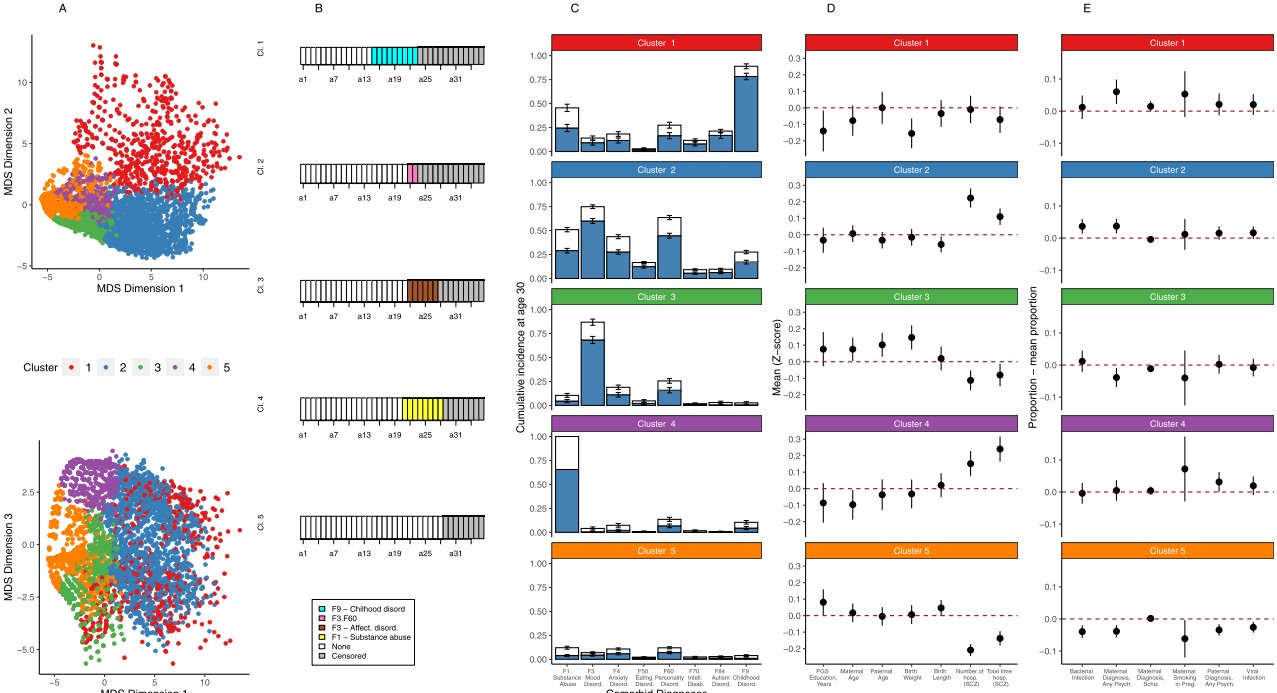

**Fig. 4 Results of the multidimensional scaling and clustering based on dissimilarity of psychiatric comorbidity sequences in schizophrenia and cumulative incidence of comorbid psychiatric disorder categories and levels of risk factors and outcomes in each of the five clusters identified.** **A** Multidimensional scaling and clustering of the 5432 individuals with schizophrenia based on the dissimilarity in psychiatric comorbidity trajectories measured by sequence analysis. **B** The most frequent state at each time interval (modal sequence) of each of the five clusters. **C** The cumulative incidence at age 30 of each comorbid diagnosis before (blue) and after (white) schizophrenia in each of the five clusters of individuals with schizophrenia. Error bars are 95% confidence intervals. Sample sizes are: Cluster 1: ($N_{none}/N_{before}/N_{after}$) F1: 321/129/103; F3: 517/50/24; F4: 490/66/33; F50: 581/<15/<6; F60: 448/92/52; F70: 514/46/20; F84: 462/98/25; F9: 66/449/59; Cluster 2: ($N_{none}/N_{before}/N_{after}$) F1: 786/426/298; F3: 411/895/204; F4: 907/416/219; F50: 1307/191/62; F60: 619/669/202; F70: 1423/85/52; F84: 1420/97/52; F9: 1146/260/144; Cluster 3: ($N_{none}/N_{before}/N_{after}$) F1: 657/28/40; F3: 118/434/112; F4: 596/75/53; F50: 700/15/16; F60: 555/109/64; F70: 717/<6/<13; F84: 713/<6/<21; F9: 716/<6/<15; Cluster 4: ($N_{none}/N_{before}/N_{after}$) F1: 0/383/201; F3: 699/<6/<31; F4: 679/15/30; F50: 723/<6/<6; F60: 631/47/42; F70: 712/<6/<11; F84: 722/<6/<11; F9: 652/30/42; Cluster 5: ($N_{none}/N_{before}/N_{after}$) F1: 1550/57/136; F3: 1675/64/44; F4: 1594/99/93; F50: 1751/23/15; F60: 1570/119/84; F70: 1733/18/24; F84: 1726/24/31; F9: 1714/16/51. **D** Mean and 95%-confidence interval of $Z$-score of continuous risk factors and outcomes for each of the clusters. Sample sizes are ($N_{cl1}/N_{cl2}/N_{cl3}/N_{cl4}/N_{cl5}$): PGS: 246/641/304/270/687; maternal age: 469/1474/702/499/1239; paternal age: 455/1453/696/493/1227; birth weight: 458/1457/684/489/1209; birth length: 578/1553/720/710/1754; number of hospitalizations and total time hospitalized: 597/1580/734/729/1792. **E** Proportion with categorical risk factor in each cluster minus the proportion in all individuals (95% confidence intervals). Sample sizes are ($N_{cl1}/N_{cl2}/N_{cl3}/N_{cl4}/N_{cl5}$): bacterial infection and viral infection: 597/1580/734/729/1792; maternal diagnosis, any psychiatric and maternal diagnosis, schizophrenia: 581/1554/726/720/1771; paternal diagnosis, any psychiatric: 580/1553/724/720/1769; maternal smoking in pregnancy: 189/412/127/94/268.

and cluster 2 (adult-disorder) (Supplementary Fig. S8), these clusters were stable (mean Jaccard coefficient > 0.59; Supplementary Table S4). The trajectory dissimilarities were thus represented well by predominantly stable and clinically interpretable groups of individuals, providing a complementary representation of clinical heterogeneity.

To test whether dissimilarities in trajectories could reflect heterogeneity in etiology, we used MANCOVAs (see "Methods" section) to relate the three principal MDS dimensions with 23 selected genetic, clinical, and environmental measures that have shown prior associations to a schizophrenia diagnosis. We additionally examined a number of hospitalizations and time spent hospitalized for psychiatric diagnoses, as commonly-used proxies of severity[38]. After Bonferroni correction for 25 tests, the three leading principal MDS dimensions associated globally with parental age ($P_{maternal\ age} = 4.6 \times 10^{-9}$; $P_{paternal\ age} = 1.9 \times 10^{-4}$), parental history of psychiatric disorders ($P_{maternal,\ any\ psychiatric} = 1.3 \times 10^{-13}$; $P_{paternal,\ any\ psychiatric} = 1.4 \times 10^{-5}$; $P_{maternal,\ schizophrenia} = 2.3 \times 10^{-4}$), birth measures ($P_{birth\ length} = 1.4 \times 10^{-7}$; $P_{birth\ weigh} = 2.8 \times 10^{-5}$), hospital treatment for infections ($P_{bacterial} = 3.3 \times 10^{-14}$; $P_{viral} = 4.7 \times 10^{-4}$), severity and long-term outcome ($P_{Number\ of\ hospitalizations} = 1.6 \times 10^{-71}$; $P_{Total\ time\ hospitalized} = 8.6 \times 10^{-37}$), and educational

attainment polygenic risk score ($P_{PGS\text{-}Education\text{-}attainment} = 1.2 \times 10^{-4}$; Table 2). Importantly, we did not see any significant association with schizophrenia polygenic risk score ($P_{PGS\text{-}Schizophrenia} = 0.18$, Table 2), which is constructed to discriminate individuals with schizophrenia from controls. In sum, multiple risk factors for schizophrenia show association with heterogeneity in comorbidity trajectory.

We then conducted post-hoc ANCOVAs to associate individual principal MDS dimensions with specific risk factors. The first principal MDS dimension (associating with higher number of comorbid diagnoses), associated positively with the two severity measures ($\beta_{Number\ of\ hospitalizations} = 2.64$, $P = 2.0 \times 10^{-29}$; $\beta_{Total\ time\ hospitalized} = 0.49$, $P = 1.5 \times 10^{-10}$), parental history of psychiatric disorders ($\beta_{maternal,\ any\ psychiatric} = 0.85$, $P = 5.8 \times 10^{-12}$; $\beta_{paternal,\ any\ psychiatric} = 0.54$, $P = 5.0 \times 10^{-5}$), maternal smoking during pregnancy ($\beta_{Maternal\ smoking\ in\ pregnancy} = 0.71$, $P = 2.2 \times 10^{-3}$), hospital treatment for bacterial infections ($\beta_{Bacterial} = 0.93$, $P = 4.7 \times 10^{-15}$), and hospital treatment for viral infections ($\beta_{Viral} = 0.54$, $P = 6.6 \times 10^{-5}$). The second principal MDS dimension (associating with comorbid childhood disorders) was associated negatively with one severity measure ($\beta_{Number\ of\ hospitalizations} = -0.65$, $P = 1.1 \times 10^{-6}$) and positively with a maternal diagnosis of schizophrenia ($\beta_{Maternal,}$

**Table 2 Results of the test for association between the principal dimensions of the comorbidity trajectories and schizophrenia risk factors and outcomes within the 5432 individuals with schizophrenia.**

| | Non-missing (N) | Joint test (MANCOVA) | | Dim 1 | | Dim 2 | | Dim 3 | |
|---|---|---|---|---|---|---|---|---|---|
| | | F | P | β | P | β | P | β | P |
| PGS - bipolar affective disorder | 2147 | 0.9 | 0.45 | | | | | | |
| PGS - depressive symptoms | 2147 | 3.0 | 0.030 | | | | | | |
| PGS - education, years | 2147 | 7.0 | $1.2 \times 10^{-4}$ | −0.21 | 0.010 | −0.10 | 0.043 | −0.14 | $1.1 \times 10^{-3}$ |
| PGS - neuroticism | 2147 | 2.6 | 0.054 | | | | | | |
| PGS - schizophrenia | 2147 | 1.6 | 0.18 | | | | | | |
| Disruptive or damaging mutations (count) | 2972 | 3.2 | 0.024 | | | | | | |
| Maternal age (years) | 4383 | 14.0 | $4.6 \times 10^{-9}$ | −0.04 | $9.8 \times 10^{-4}$ | −0.02 | $4.8 \times 10^{-3}$ | −0.03 | $1.5 \times 10^{-6}$ |
| Paternal age (years) | 4324 | 6.6 | $1.9 \times 10^{-4}$ | −0.02 | 0.014 | −0.01 | 0.26 | −0.02 | $3.0 \times 10^{-4}$ |
| Gestational age (days) | 4311 | 2.8 | 0.040 | | | | | | |
| Birth length (cm) | 5315 | 11.6 | $1.4 \times 10^{-7}$ | −0.05 | 0.013 | −0.02 | 0.16 | −0.05 | $1.7 \times 10^{-7}$ |
| Birth weight (percentile for GA) | 4297 | 7.9 | $2.8 \times 10^{-5}$ | −0.43 | 0.034 | −0.30 | $9.8 \times 10^{-3}$ | −0.37 | $4.4 \times 10^{-4}$ |
| APGAR 5 | 5370 | 2.1 | 0.10 | | | | | | |
| Maternal smoking in pregnancy | 1090 | 5.3 | $1.3 \times 10^{-3}$ | 0.71 | $2.2 \times 10^{-3}$ | 0.10 | 0.55 | 0.29 | 0.020 |
| Bacterial infection | 5432 | 22.1 | $3.3 \times 10^{-14}$ | 0.93 | $4.7 \times 10^{-15}$ | 0.00 | 0.97 | 0.17 | $8.2 \times 10^{-3}$ |
| Viral infection | 5432 | 6.0 | $4.7 \times 10^{-4}$ | 0.54 | $6.6 \times 10^{-5}$ | 0.10 | 0.22 | 0.07 | 0.35 |
| CNS infection | 5432 | 0.4 | 0.79 | | | | | | |
| Otitis infection | 5432 | 4.4 | $4.4 \times 10^{-3}$ | | | | | | |
| Maternal infection during pregnancy, bacterial | 5432 | 3.0 | 0.028 | | | | | | |
| Maternal infection during pregnancy, viral | 5432 | 2.2 | 0.084 | | | | | | |
| Maternal, any psychiatric | 5352 | 21.1 | $1.3 \times 10^{-13}$ | 0.85 | $5.8 \times 10^{-12}$ | 0.18 | 0.014 | 0.22 | $5.3 \times 10^{-4}$ |
| Maternal, schizophrenia | 5352 | 6.5 | $2.3 \times 10^{-4}$ | 0.23 | 0.49 | 0.77 | $6.7 \times 10^{-5}$ | 0.27 | 0.12 |
| Paternal, any psychiatric | 5346 | 8.4 | $1.4 \times 10^{-5}$ | 0.54 | $5.0 \times 10^{-5}$ | 0.06 | 0.42 | 0.21 | $2.0 \times 10^{-3}$ |
| Paternal diagnosis, schizophrenia | 5346 | 4.4 | $4.3 \times 10^{-3}$ | | | | | | |
| Number of hospitalizations (SCZ) | 5432 | 113.9 | $1.6 \times 10^{-71}$ | 2.64 | $2.0 \times 10^{-29}$ | −0.65 | $1.1 \times 10^{-6}$ | 1.71 | $5.2 \times 10^{-46}$ |
| Total time hospitalized (SCZ) | 5432 | 57.8 | $8.6 \times 10^{-37}$ | 0.49 | $1.5 \times 10^{-10}$ | −0.12 | $3.7 \times 10^{-3}$ | 0.45 | $9.5 \times 10^{-31}$ |

Non-missing (N) indicates the number of individuals for whom data were available; for details on the distribution of the variables see Supplementary Table S7 and Supplementary Figure S9. Joint test is the results of MANCOVAs done with the first three principal MDS dimensions as a dependent variable and one degree of freedom. For variables with P-values significant after adjustment for 25 tests ($P < 0.002$), post-hoc tests were performed. β is the β-coefficient from the post-hoc linear regressions with each of the dimension scores as a dependent variable, the adjacent P-values are the results of ANOVA F-test with one degree of freedom. P-values shown are uncorrected. All analyses were adjusted for age and sex. PGS-Educational-attainment is adjusted additionally for 10 principal components and genotype wave.

schizophrenia = 0.77, $P = 6.7 \times 10^{-5}$). The third dimension (associating with comorbid substance use disorders) showed positive associations with severity measures ($\beta_{\text{Number of hospitalizations}} = 1.71$, $P = 5.2 \times 10^{-46}$; $\beta_{\text{Total hospitalization time}} = 0.45$, $P = 9.5 \times 10^{-31}$), and parental psychiatric diagnoses ($\beta_{\text{maternal, any psychiatric}} = 0.22$, $P = 5.3 \times 10^{-4}$; $\beta_{\text{paternal, any psychiatric}} = 0.21$, $P = 2.0 \times 10^{-3}$), and negative associations with birth length ($\beta_{\text{birth length}} = -0.05$, $P = 1.7 \times 10^{-7}$), birth weight ($\beta_{\text{birth weight}} = -0.37$, $P = 4.4 \times 10^{-4}$), with parental age ($\beta_{\text{maternal age}} = -0.03$, $P = 1.5 \times 10^{-6}$; $\beta_{\text{paternal age}} = -0.02$, $P = 3.0 \times 10^{-4}$), and with PGS for educational attainment ($\beta_{\text{PGS-Educational-attainment}} = -0.14$, $P = 1.1 \times 10^{-3}$, Table 2). This shows that the three principal MDS dimensions had distinct patterns of associations with risk factors.

We paralleled these dimensional analyses using membership in the five clusters as outcomes and risk factors as predictors. We visualize the mean and prevalence for the 13 risk factors that were significant in the dimensional MANCOVA for each of the clusters (Fig. 4D, E). We then used multinomial logistic regression to show global associations ($P < 0.002$) with membership in the five clusters for 8 of these 13 variables. Post-hoc single cluster-single risk factor associations were significant ($P < 0.00096$) in 12 out of 52 tests and showed patterns broadly concordant with the post-hoc dimensional analysis (Supplementary Table S8). We note that this clustering, while providing a potentially more intuitive representation of clinical heterogeneity, is a reduced information view and by forcing a categorical measure onto a dimensional

phenomenon, may lose sensitivity. When considering clusters and risk factors, we saw that some (5 out of 13) associations found with the dimensional representation of the trajectories were not significantly associated with cluster membership after a Bonferroni correction ($P < 0.002$). We believe this is due to a lack of sensitivity induced by the clustering and note that all 13 associations were at least nominally significant ($P < 0.05$). Thus, the intuitive factor loadings in the dimensions were also largely reflected in the five clusters. In order to enable external replication and use of these clusters, we trained a decision tree on the $k = 5$ MDS-based clustering that produces a largely overlapping classification (rand index = 0.83; Supplementary Fig. S8) with highly concordant associations to risk factors and outcomes (Supplementary Fig. S9).

Analyses of the sensitivity to imputation, substitution cost, and alignment method settings of the SA suggested that the results are stable and robust (Supplementary Note 6 and Supplementary Table S11). Further, the change from ICD-8 to ICD-10 on Jan 1, 1994, the addition of outpatient contacts in 1995, and changes in diagnostic practice over time more generally could result in birth cohort effects. To address this, we calculated the cumulative incidence of each comorbidity stratified by birth year. We observed the cumulative incidence of comorbid mood disorders, autism spectrum disorders, and childhood disorders increase with birth year while comorbid personality disorders decrease (for a more detailed discussion of possible cohort effects see Supplementary Note 7 and

Supplementary Fig. S11). Our sensitivity analyses suggest, however, that this cannot account for our results (Supplementary Notes 6 and 7). Additionally, we sought replication in a smaller independent sample of individuals ($N = 870$) ascertained from other iPSYCH-cohorts[39] and diagnosed with schizophrenia after 2012. This cohort does not have the same population representativeness of the initial cohort due to its ascertainment which should lead to under-estimates of the robustness of our initial findings. Regardless, we found sign concordance for 12 of the 13 univariate dimensional associations (Supplementary Table S10) and the results were strictly significant for five (educational attainment PGS ($P = 0.0085$), maternal age ($P = 0.0003$), maternal smoking during pregnancy ($P = 0.0007$), number of hospitalizations ($P = 0.0002$) and total time hospitalized ($P = 0.0004$, see Supplementary Table S9)). Taken together, extensive sensitivity analysis and replication suggest the associations between clinical heterogeneity and etiological factors are stable and reproducible.

## Discussion

Here, we have leveraged nationwide, population-based hospital registers to describe the structure of inter-individual differences in longitudinal trajectories of comorbid psychiatric diagnoses for a complete birth cohort of individuals diagnosed with schizophrenia. Our report shows that four out of five individuals in Denmark with schizophrenia are diagnosed with at least one other major psychiatric disorder during the period from birth to age 36 years and the vast majority of these occur prior to the first diagnosis of schizophrenia. Importantly, there is structure across our follow-up period, in that particular temporally ordered pairs of comorbidities occur more frequently (e.g., individuals with personality disorders have higher risk of subsequent substance abuse disorders). We compared the complete trajectories across the entire follow-up period and identified three latent principle dimensions that explained a plurality of the variance. These three dimensions clustered individuals with schizophrenia into five stable subgroups with intuitive, interpretable, and clinically rele-vant factor loadings and revealed distinct patterns of associations with schizophrenia risk factors. Thus, our exploratory and data-driven analyses revealed substantial, stable disorder-course het-erogeneity among individuals with schizophrenia, which could potentially be rooted in etiological differences.

There are many reasons an individual may receive a particular diagnosis prior to schizophrenia but not all would result in diagnoses that are necessarily meaningful for considering sub-groups. We found that 734 individuals had a trajectory in cluster 3 characterized by affective disorder diagnoses, mostly occurring before the onset of schizophrenia. However, this cluster had low stability and only a few of the putative risk factors showed sig-nificant associations with Cluster 3, which could indicate that comorbid affective disorder diagnosis alone does not capture any specific pathology. Notably, prodromal symptoms of schizo-phrenia can overlap with depression symptoms[40]. In contrast to this, clusters 1, 2, 4, and 5 were fairly stable and most of the associations with risk factors and outcomes seen in dimensional representation were also associated with clusters 1–3 when comparing to cluster 5. This indicates that features loading on these clusters (childhood disorders, multiple adult comorbidities, and substance abuse) form more distinct trajectories and risk factor and outcome profiles. This could be consistent with childhood disorders and substance abuse having more symptoms clearly distinguishable from typical schizophrenia symptoms, marking more stable heterogeneity, while a prior depression diagnosis may be more epiphenomenal (e.g., relating to pro-dromal symptoms or particularities of clinical practice).

Our work may add additional context to well-described case-control discriminating risk factors[41]. For example, dimension 1, reflecting a disorder-course characterized by multiple co-occurring psychiatric diagnoses, was particularly associated with parental history of mental disorders, lower parental age at birth, maternal smoking during pregnancy, birth complications, a his-tory with a higher load of hospital-treated infections, and a more severe disorder course as captured by both number of hospitali-zations and total time hospitalized for their psychiatric illness. There has been much work previously on infections being a risk factor for schizophrenia[42], which have previously been shown to increase the risk of schizophrenia in a dose-response relationship with the number of severe infections[43], that did not appear to be confounded by the genetic risk for schizophrenia[44]. Our work suggests that beyond a diagnosis, infections may predispose to a disorder course. Along these same lines, we also find associations for risk factors that have more mixed support in the current literature, such as parental age[45,46]. Our work may add clarity to this, suggesting the effects of risk factors are not uniform throughout the population of individuals with schizophrenia and may only associate with certain segments of the population. Other associations did not seem supported by previous literature e.g., the finding that family history of mental disorder is associated with a disease course with multiple psychiatric comorbidities.

This work is also relevant to the position of schizophrenia as a neurodevelopmental disorder[47]. The second dimension, which was, in particular, capturing clinical trajectories denoted by increased prevalence of comorbid childhood psychiatric diag-noses could reflect the predominance of an earlier developmental pathway that presents in childhood, possibly overlapping with what have been described as premorbid schizophrenia symptoms[47]. This is in line with a recent study by Dickinson et al.[48] who used a cross-sectional design and found a sub-group of individuals with schizophrenia with signs of a more neuro-developmental course. This dimension was associated with a maternal schizophrenia diagnosis. Notably, the pioneering work by Fish et al.[49] described the developmental pandysmaturation syndrome among newborns of mothers with schizophrenia. This dimension was associated with a less severe disease course after the schizophrenia diagnosis as measured by the number of admissions. Interestingly, Dickinson et al.[48] also found that individuals with schizophrenia and a pre-adolescent cognitive impairment had less severe symptoms after onset, than those with more typical pre-adolescent cognitive performance that declined (adolescent disruption of cognitive development). Both dimen-sions 1 and 2 could co-adhere with subtypes of schizophrenia with more neurodevelopmental components, where particularly dimension 1 has a higher level of early life risk factors for schi-zophrenia, which could indicate that neurodevelopmental pathology plays a particularly important role for these dimensions.

Dimension 3 was primarily related to comorbid substance use disorders. This was associated with a more severe disorder course as measured by the number of hospitalizations and also more parental psychiatric diagnosis, increased birth complications, younger parental age at birth, and lower PGS for educational attainment. When taken together, these risk factors could suppose a less resilient or robust environment for the individual, in that birth complications, parental age, and education level may be correlated with socio-economic status and access to support[50]. Comorbid substance use could also lead to a more severe, pro-longed course, by limiting the ability to obtain and maintain treatment[51]. Thus, this dimension could represent the presenta-tion of individuals who express schizophrenia within the context of a challenging environment.

All three dimensions showed some associations with variables that are known to be correlated with socio-economic factors, such as income level and parental education, which may affect the home environment (e.g., parental age, maternal smoking during pregnancy, educational-attainment-PGS, parental psychiatric disorders). However, the direction of causality is difficult to infer[52–54]. While socio-economic factors and the home environment may directly influence disorder course (e.g., Wimberley et al.[55]), any genetic factors associated with specific, life-trajectories of schizophrenia could alternatively affect the home environment. This second scenario could occur if a more debilitating form of the disorder leads to more home disruption or if parents, who by way of Mendelian inheritance would carry these same genes, expressed any behavioral changes that affect the home environment. This kind of gene-environment correlation across generations can be tested by studying non-transmitted alleles as has been done for educational attainment[56]. We should note that in this study, relationships between environmental risk factors and family history of psychiatric disorders are not completely confounded, implying that environmental factors are likely to make independent contributions to disorder course. The notion of independent effects of socio-economic factors and parental history of psychiatric disorders has found previous support in contributing to risk for schizophrenia[54], and provide an interesting motivation for future gene by environment or causal inference studies of heterogeneity in outcomes and disorder course for individuals with psychiatric disorders. Given the current weight of evidence, it is more likely that the home environment modifies disease course, rather than the other way around, although more work is needed, especially given the emerging perspectives of cross-generation generational genetic effects confounding intuitive causal relationships.

Our initial motivation for pursuing trajectory analysis in schizophrenia was to identify heterogeneity in factors that precede the first diagnosis of schizophrenia. However, we observed many diagnoses of childhood psychiatric disorders (intuitive predecessors) recorded after the first diagnosis of schizophrenia, such that they appear in adolescence or adulthood. Although counterintuitive, earlier onset disorders may go undetected until deeper psychiatric examinations are prompted by treatment for other indications or that a large portion of care for these has been administered at primary care facilities not captured by the registers. The abundance of these highlights the importance of studying for example the safety of administering stimulants to individuals with a history of psychosis[57]. However, the nature of these later-recorded childhood disorders requires more targeted follow-up.

While polygenic scores for educational attainment and schizophrenia both discriminate schizophrenia cases from controls in this sample, only the educational attainment score was associated with the structure of differences in comorbidity trajectories among individuals with schizophrenia. Relatively few studies[14,20,58] have pursued analyses that do not focus on the primary disorder polygenic score, a common design (e.g., refs. [59,60]). While intuitive, there is not a theoretical requirement to believe case-control associated variants are the only relevant predictors of heterogeneity in outcomes. In fact, while some studies found SCZ-PGS to be related to chronicity[59] and negative or disorganized symptoms[14,20], several well-powered studies have been unable to find primary disorder-related PGS to be associated with clinical heterogeneity including treatment response in schizophrenia[60] and age of onset in bipolar disorder[61]. We find it particularly interesting that PGS for educational attainment best discriminate individuals with different trajectories, as reports show educational attainment has large genetic overlap (i.e., shares associated loci) with schizophrenia, but without consistent correlation in the direction of the per locus effects (i.e., genetic correlation estimates near 0)[62]. In Frei et al.[62], the authors demonstrated that co-occurring associations without evidence of genetic correlation can be consistent with etiological heterogeneity, a hypothesis that finds support in this work. Additionally, a recent study by Dwyer et al.[16] characterizing symptom trajectories after the onset of psychosis also found an association with educational attainment PGS which further supports our finding that genetic variants linked to educational attainment are associated with longitudinal heterogeneity.

The structured, stable variability in clinical course among individuals with schizophrenia, and its associations to possible etiological factors, as indicated by our MDS analyses have at least three possible implications for clinical care. First, the principal MDS dimensions of the comorbidity trajectories were differently associated with aspects of the downstream need of care, e.g., number of hospitalization and total time hospitalized, suggesting sub-group predictors could have implications for planning and prioritizing healthcare among those most in need. Second, many of the etiological factors associated with clinical course captured by the comorbidity trajectories were early life factors (e.g., genetic risk scores, birth variables, parental history), suggesting it could be possible to predict the expected clinical course at early stages (e.g., first psychotic episode). Finally, psychiatric classifications are continually updating and could be further refined on the basis of stable differences in outcomes and disorder course having implications for nosology and diagnostic practice more broadly.

The strengths of this study are the population-wide design that encompasses all individuals with schizophrenia in the birth cohort at the time of ascertainment, the use of complete longitudinal records of all hospital-assigned diagnoses, and admissions over 16-36 years of follow up, and an exploration of potential etiology heterogeneity.

The present trajectory-based method is limited by the relatively young age of the population birth cohort (1981–2002) where some individuals may still go on to develop schizophrenia or other psychiatric disorders and including these additional late-diagnosed individuals may add or change the representations of the structure in trajectory variability. For example, female individuals with schizophrenia may have later onset[63] and different comorbidity profiles[8], and thus early-onset trajectories might less dominate the overall structure. As such this study can be viewed as a minimal summary of the structure within the variability among the course of schizophrenia and a first step towards more thorough characterizations of the nature of underlying heterogeneity.

The use of psychiatric diagnoses to describe symptom presentation is subject to a number of limitations including imperfect inter-rater reliability and the reliance on imperfect classification systems. Further, the study could be viewed as somewhat limited by the use of health register diagnoses as opposed to standardized research-based diagnoses. Registers provide complete coverage of public hospital in- and outpatient diagnoses, but treatments by private psychiatrists or general practitioners are not recorded[23], and it has been found that substance use disorders in admissions for schizophrenia were underdiagnosed[64]. This limitation may be mitigated by the notion that more severe cases, such as those diagnosed with schizophrenia, tend to be treated within the hospital system[23] and typically have multiple hospital contacts such that the register coverage of comorbidities might be better in these clinical populations. Numerous prior studies have shown the overall diagnoses in the registers, including for schizophrenia, to be reliable[23,25–28]. This may reflect that registered diagnoses are consensus assignments made at the end of hospitalization by the college of specialized psychiatrists and based on the full clinical course and set of examinations, treatments, and

outcomes. Although we see evidence for the change in diagnostic system from ICD8 to ICD10 occurring in 1994 in the rates of specific comorbidities among individuals with schizophrenia, our specific results appear robust to this effect. However, in the interpretation of the parental history, it should be kept in mind that while the majority of diagnoses in the probands were assigned under ICD-10, many of the parental diagnoses were assigned under ICD-8, which does not completely overlap with current diagnostic criteria[65]. Future extensions of this work should still be mindful of potential complications.

Another potential limitation is that we cluster individuals according to changes of diagnostic states in 1-year windows, rather than using finer-grained symptom scales in shorter time intervals (e.g., Kotov et al.[15] or Dwyer et al.[16]). We acknowledge that diagnoses may not capture the full nuance of a change in clinical presentation over time and thus think our results speak to how etiological differences may contribute to longer-term, life-course presentations. We feel this is still important for con-ceptualizing how to manage long-term care and is somewhat complementary to investigations of symptom scales on short timelines. Further, the potential limits of cruder diagnostic changes may be balanced by the strengths of our unique data resource—long follow up, population-wide, birth cohort—that are not currently mirrored in cohorts with deeper symptom-scale phenotyping. Developing such cohorts would be highly infor-mative and complement our work here. While we focused on detecting and describing heterogeneity within schizophrenia, more studies are needed to uncover how this relates to other diagnostic categories, for example, broadening to a wider spec-trum of disorders involving psychosis, and our results should be replicated externally, preferably using cohorts from another country.

In addition, the analyses of polygenic risk scores are limited by the study design and phenotype definitions in the discovery GWAS (e.g., other factors than biological intelligence may impact the performance in intelligence tests).

Finally, we acknowledge that we have surveyed only a very limited set of potential disorder-course altering risk factors: aggregate measures of common SNPs instantiated as polygenic risk scores (PGS), rare single nucleotide variants, and clinical variables with prior support, but we feel that more far-reaching searches, including other measures of the social environment (e.g., household income or parental education or marital status), copy number and structure variants, broader collections of clin-ical factors, and PGS from multiple traits and diseases are needed to more fully characterize the nature of disorder-course hetero-geneity in schizophrenia. In addition, future work could emphasize longer-term outcomes, such as participation in the labor market.

## Methods

**Participants**. A cohort consisting of all singleton births by a known mother in Denmark between May 1, 1981 and December 31, 2002 and diagnosed with schizophrenia before December 31, 2012, was identified through the Danish Psy-chiatric Central Research Register (PCRR)[23]. We included all individuals with an inpatient or outpatient hospital contact discharge code corresponding to schizo-phrenia in the International Statistical Classification of Diseases and Related Health Problems 10th revision (ICD-10) codes F20.0-F20.9.

**Data sources**. Using the unique Danish personal registration number (CPR), data on all psychiatric diagnoses assigned before December 31, 2016 were obtained from The Danish National Patient Register (DNPR)[22] and the PCRR[23]. The PCRR contains data on diagnoses given during hospital admission and from 1995 onwards it also contains diagnoses given at outpatient clinics. For every individual with a schizophrenia diagnosis and 10,864 age and sex-matched population con-trols (Supplementary Note 1), we obtained the date of the first diagnosis for a broad selection of diagnoses of ICD-10-F chapters 1 or 3–9 (assigned after Jan 1, 1994) or a corresponding[66] ICD-8 diagnosis (assigned before Jan 1, 1994; Supplementary

Table S1; ICD-9 was not implemented in Denmark). F43.2–48—the etiologically-defined stress disorders—were omitted. Additionally, the number of psychiatric hospitalizations and days hospitalized with a schizophrenia diagnosis. Since at the time of the study no private psychiatric hospitals existed in Denmark, the registries are considered practically complete for individuals diagnosed with severe psy-chiatric disorders such as schizophrenia[23]. The Danish Neonatal Screening Bio-bank contains dried blood spots (DBS) for almost every Dane born after 1981[67]. The quality of amplified DNA (aDNA) from DBS samples for genotype analyses has been shown to be equivalent to high-quality DNA samples[68]. The sample consisted of two sub-cohorts: The GEMS cohort (Genomic Medicine for Schizophrenia[69]), born 1981–1996; and a subset of the iPSYCH cohort[39], born 1981–2002 and not included in the GEMS cohort. aDNA genotyping using DBS samples has been performed on both cohorts but using different chips (Illumina 610k and Illumina PsychArray). In the present analyses, all individuals with a schizophrenia diagnosis were included in the phenotypic characterization, but only the iPSYCH cohort was used in genetic association analyses to avoid batch effects. Quality control procedures for genotype data including procedures to account for genetic ancestry have been described elsewhere[19]. Briefly, the Infinium PsychChip v1.0 array was used for amplified DNA obtained from dried blood spots. Of the ~550,000 genotyped SNPs, 246,369 were deemed good quality, these were phased using SHAPEIT3[70], and imputed with Impute2[71] using the 1 000 genomes project phase3[72] as reference. Imputed genotypes were filtered on imputation quality (INFO > 0.2), association with imputation batch ($P > 5 \times 10^{-8}$), association with genotyping wave ($P > 5 \times 10^{-8}$), Hardy–Weinberg equilibrium (HWE; $P > 1 \times 10^{-6}$), differing imputation quality between cases and controls ($P > 1 \times 10^{-6}$), and minor allele frequency (MAF > 0.01). After these steps, 8,019,760 variants remained. EIGENSOFT v6.0.1[73], was used to select individuals of homogeneous genetic. KING v1.9[74] was used to estimate kinship and from each pair of patients with closer than third-degree kinship one was removed. No samples had high levels of missing genotypes (>1%), abnormal heterozygosity, or genotype/recorded sex discordance. For a subset of individuals, whole-exome sequencing (WES) was performed. Quality control and count of disruptive or damaging mutations were calculated from exomes according to the definitions in Ganna et al[75]. In brief, the WES was performed using the Illumina nextera Rapid Capture Exome kit, with a mean coverage of 77×. Mapping and variant calling followed the Broad Institute Pipeline[75]. Samples were excluded based on contamination, sex-mismatch, non-European ethnicity, and genetic relatedness[75]. Variants were excluded based on read depth (<20), HWE ($P < 0.000001$) or call rate (<0.80). Protein truncating variants were, classified using VEP version 85. Family history of psychiatric dis-orders was obtained from the PCRR using the parental CPR obtained from the Danish Civil Registration System[24]. Family history of schizophrenia was defined as ICD-8 (before Jan 1, 1994) codes 295.x9 excluding 295.79; ICD-10 (after Jan 1, 1994) codes F20.0-F20.9. Similarly, a history of maternal infection during preg-nancy was obtained from DNPR[22] and birth-related variables were obtained from the Danish Medical Birth Register[76] (for details see Supplementary Table S7).

The Danish Scientific Ethics Committee, the Danish Health Data Authority, the Danish data protection agency, and the Danish Neonatal Screening Biobank Steering Committee approved this study. This is in keeping with the strict ethical framework and the Danish legislation protecting the use of these samples. At the time of blood sampling, parents were informed in writing that the blood spots are stored in the Danish Neonatal Screening Biobank and can be used for research, pending approval from relevant authorities and about how to prevent or withdraw the sample from inclusion in research studies[39]. The use of this data is considered to be in accordance with the WMA Declaration of Taipei[77].

**Replication sample**. An additional cohort consisting of individuals diagnosed with schizophrenia between January 1, 2013 and December 31, 2016 was obtained using individuals that were included in the iPSYCH study[39] as part of either a different sub-cohort (i.e., with a diagnosis of attention deficit hyperactivity disorder, autism spectrum disorders, or mood disorders, but not schizophrenia, before December 31, 2012) or as part of the random population sample. This independent cohort was reserved for replication of results.

**Cumulative incidence and survival analysis**. Cumulative incidence was com-puted separately for each of the eight comorbid disorder categories. We considered an individual censored on the date of whichever came first of emigration ($N = 65$) or death ($N = 154$) according to the civil registration system or at the end of follow-up (Dec 31, 2016). Within the primary cohort diagnosed with schizophrenia before December 31, 2012, we computed the hazard ratio of getting a diagnosis from each of the eight comorbidity categories given each of the other seven cate-gories, using a Cox Proportional Hazard model, defining censoring as described above, with the first diagnosis as a time-dependent covariate and adjusting for age and sex. Apart from the two diagnoses being modeled, other diagnoses were not included in the model.

**Sequence analysis**. Sequence analysis is ideally suited to the complex nature of these data, in that it can measure dissimilarities among trajectories constructed with multiple (8) categorical outcomes[33]. Each individuals psychiatric medical history was transformed into a sequence of states, wherein each state corresponded

to a period of 1 year, from birth to the end of 2016 (Fig. 3). The sequence alphabet consisted of a no diagnosis-state, a state for selected diagnoses from each of the eight diagnostic chapters (Supplementary Table S1), and a state for each of the 247 possible combinations of diagnoses. For each year until the end of follow-up, an individual's state was defined as the cumulative set of diagnoses they had received before the beginning of that year.

Optimal matching[78] was used to calculate dissimilarities between pairs of sequences. This method aligns sequences using state substitutions, insertions, and deletions (indels). As sequences had varying observation lengths (16–35 years) depending on the year of birth, for all sequences, we computed the probabilities of later states up to length 36 using a first-order Markov chain based on population-wide estimates of transition probabilities (Supplementary Note 3). As substitution costs in optimal matching, we used the Jaccard distance of the diagnoses contained in the states (e.g., F1 to F3 costs 1, F1-F3 to F4-F5 costs 1, F1-F3 to F3-F4 costs 0.5, etc.; Supplementary Table S3). Indels were set to the unit cost. For the censored states, the cost was the weighted mean of the costs of all possible substitutions the weighting being the probability of being in that state (Supplementary Note 3). In this fashion, we obtained a dissimilarity metric that did not depend on the length of the observed sequences and with the properties of a Euclidean distance, suitable for application to classical multidimensional scaling (MDS).

**Multidimensional scaling and clustering**. Based on the calculated dissimilarities, a classic MDS was performed. We calculated the cumulative $R^2$ of MDS with the number of dimensions ($k$) varying from 1 to 13. Additionally, we calculated stress plots for each value of $k$. Stability of the MDS results was assessed using a permutation test. Scores for each retained dimension were obtained for every individual. To bring the dissimilarities to a more easily interpretable form (i.e., groups of individuals with similar trajectories), we used the scores from the MDS to perform a Ward's hierarchical agglomerative clustering[37] and selected the number of clusters where the proportion of variance explained leveled off. A permutation test of the stability of the clustering was undertaken.

**Associations with clinical and genetic risk factors**. Multivariate analysis of covariance (MANCOVA) was used to test for association between selected MDS dimension scores of dimensions 1–3 (dependent variables) and 18 putative risk variables from the registries, including pregnancy and birth-related variables, hospital contacts due to an infection both in the individual and maternal infections during the pregnancy period, and family history of psychiatric disorders (Supplementary Table S7). MANCOVAs included covariates for age at the end of the study period and sex.

Similarly, we tested for associations with a selection of psychiatric and cognitive polygenic scores (PGS) and count of rare deleterious or damaging mutations. PGS were calculated by using genetic loci previously found associated with a trait. For each locus, an individual can have 0, 1, or 2 risk alleles. The PGS is the sum of risk alleles carried by an individual weighted by the effect size of that allele in the original study. Seven PGSs from publicly available summary statistics of published GWASs were computed (Supplementary Table S5 and Supplementary Fig. S9). These included studies of bipolar disorder[79], depressive symptoms[80], educational attainment[81], extraversion[82], performance in intelligence tests[83], neuroticism[80], and schizophrenia[84]. Of these, five were found to be associated with schizophrenia when compared to a random population cohort (Supplementary Table S6). These five PGSs and the count of rare mutations were used in MANCOVAs with MDS dimensions (1–3) as dependent variables while adjusting for the first ten principal components (PCs) of genetic ancestry, genotyping wave, age, and sex.

If a given MANCOVA showed a significant association with the independent variable after a Bonferroni correction for the total number of MANCOVA $F$-tests conducted ($P < 0.05/25 = 0.002$), we performed post-hoc univariate regressions to determine which particular dimension was associated with the variable.

A multinomial logistic regression was conducted to test for associations between the variables with a significant association in the MANCOVA and the cluster membership in the selected ($k = 5$) clustering. These were conducted sequentially with the cluster membership as a dependent variable and each of the predictor variables as an independent variable with adjustment for sex and age and for genetic variables additional adjustment for 10 PCs and genotyping wave. Odds ratios were reported treating cluster 5 as reference. Since these were selected based on association in the MANCOVA, we conservatively applied the same significance threshold ($P < 0.05/25 = 0.002$).

**Associations with outcome**. Additionally, we tested whether disease trajectories described by the selected MDS dimensions (1, 2, and 3) were associated with the average yearly number of hospital admissions under a schizophrenia diagnosis after the first diagnosis of schizophrenia and the average number of days spent in hospital per year. For these analyses, we used MANCOVAs as described above adjusting for age and sex.

**Replication**. All significant associations from the MANCOVAs were tested in an independent replication cohort consisting of individuals born within the same years but diagnosed with schizophrenia between January 1, 2013 and December 31, 2016. Using the same SA dissimilarity metric, these individuals were projected onto the MDS dimensions of the primary cohort.

**Sensitivity analyses**. To assess the robustness of the SA findings, we repeated the analyses within a sub-cohort which was obtained by selecting all sequences with a length of exactly 25 (i.e., only individuals born before January 1, 1991, diagnosed with schizophrenia before age 25 and including only their comorbidity trajectories up until age 25). Dissimilarities were then computed without imputation and using three different substitution cost schemes, three different indel costs, and two measures of distance between state distributions[78] (Supplementary Note 6).

**Reporting summary**. Further information on research design is available in the Nature Research Reporting Summary linked to this article.

## Data availability

We provide an exploratory web-portal at https://diagtraj.shinyapps.io/diagtraj/ that enable the reader to explore MDS dimension 4–7 and a higher number of clusters. In accordance with the consent structure of iPSYCH and Danish law, individual-level genotype and phenotype data are not able to be shared publicly. However, all relevant intermediate-level data can be made available on request and following appropriate ethical review. Source data are provided with this paper.

## Code availability

R code is available at https://github.com/MortenKrebs/Trajectories_in_schizophrenia[85].

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

## Acknowledgements

This work is funded by the Lundbeck Foundation (R165-2013-15320, R102-A9118, R155-2014-1724, R248-2017-2003 (iPSYCH), and R230-2016-3565 (M.D.K)) and by National Institute of Health (R01MH124789-01).

## Author contributions

M.D.K., T.W., and W.K.T. designed the study. T.W., O.M, A.D.B., D.H., P.B.M., and M.N. collected the data. M.D.K., G.E.T., and W.K.T. conducted the statistical analyses. M.B., M.G., D.H.G., and C.C.F. aided in interpreting the results. M.D.K., A.J.S., W.K.T., and T.W. wrote the initial draft. All authors revised and approved the final manuscript.

## Competing interests

The authors declare no competing interests.
