## [Peer Review File · Nature Communications]

Associations between patterns in comorbid diagnostic trajectories of individuals with schizophrenia and etiological factorsREVIEWER COMMENTS

Reviewer #1 (Remarks to the Author):

In this interesting study, the authors sought to explore the heterogeneity of schizophrenia, using longitudinal, population-based birth cohort data collected through the Danish health registries on over 5k individuals diagnosed with schizophrenia. Individual "trajectories" were reconstructed based on ages at onset and sequence of comorbid diagnoses.

Over 80% of cases had one or more additional psychiatric diagnosis. This comorbidity was not random; rather, certain diagnoses were assigned to the same patient more or less often over time. For example, mood disorders were associated with increased hazard of subsequent personality, eating disorders, anxiety, and obsessive-compulsive disorders. Multidimensional scaling found 3 factors that explained almost 80% of the variance. Logistic regression analysis of MDS scores showed that each of the factors was associated primarily with one variable: number of co-occurring diagnoses, child vs adult onsets, and substance abuse. Cluster analysis of MDS scores found 5 case groups that accounted for ~50% of the variance in trajectory. Post-hoc ANOVAs showed that the MDS scores (and clusters) showed distinguishable patterns of association with parental age, parental psychiatric diagnoses, birth weight and length, hospital treatment for infections, hospitalization frequency, and a polygenic score (PGS) weighted for educational attainment, but no significant association with PGS weighted for schizophrenia or other major psychiatric disorders. Replication testing in other iPSCYH cohorts showed concordant trends. The authors conclude that their findings replicate previous associations between schizophrenia and infections, and identify "new likely causal relationships", suggesting that etiological heterogeneity alters the presentation and course of schizophrenia.

The main strength of this study is the sample, which is large, longitudinal, and representative of the population studied. The analysis approaches are fairly standard, albeit largely exploratory, and apparently sound. Support is found for some of the findings in other samples, but mainly as directional trends rather than significant findings. Many of the findings are consistent with past epidemiologic studies showing that schizophrenia, like other mental health diagnoses, is usually comorbid with others; and that paternal age, maternal health, prenatal growth and nutrition, and childhood infections contribute to risk for schizophrenia. The case clusterings based on patterns of comorbidity are somewhat new, but largely recapitulate long-known distinctions between affective and non-affective psychoses, earlier onset of illness, and the grave impact of substance abuse both before and after diagnosis. The largely negative associations with psychiatric PGS are a great disappointment, but may reflect limitations inherent in the original case-control samples on which the PGS weights are based. Lacking strong PGS or other genetic findings, however, I don't think the conclusion that "new likely causal relationships" have been identified is well supported.

Reviewer #2 (Remarks to the Author):

NCOMMS-20-16791

This study aimed to explore inter-individual patterns in comorbid diagnostic trajectories in schizophrenia and other non-affective psychotic disorders. Overall, this is a well written manuscript, it is well structured, and it clearly articulates its contribution to our understanding of the heterogeneity of clinical trajectories of non-affective psychotic disorders. Overall, the paper is of substantial interest, but could be improved further by addressing the following points, which are listed in approximate order of appearance in the manuscript.

1. This study focused on schizophrenia and other non-affective psychotic disorders (ICD-10, F2) – what was the reason for ignoring affective psychotic disorders (ICD-10, F3, psychotic codings) given there is substantial overlap in genetic and socio-environmental risk among affective and non-affective psychoses (Cross-Disorder Working Group of the PGC, 2013)? To refer to ICD-10 F2 as diagnosis of 'schizophrenia' is, strictly speaking, wrong, as several other non-affective psychoses are grouped in this category. This is somewhat surprising to read in a paper intending to focus on clinical heterogeneity and comorbidity.

2. Given the large sample size of the primary naturalistic cohort, it remains unclear why this was not used to generate two random samples for cross-validation to ensure findings observed in an estimation sample are directly replicated in a (statistically independent) validation sample rather than in a non-equivalent cohort recruited in a different time frame. Please clarify.
3. The diagnostic states defined remain rather crude, as each state corresponds to a period of one year. Clinical states of people with psychosis can vary markedly throughout the course of a year. There is also substantial evidence that the heterogeneity of schizophrenia and other psychoses is accounted for by symptom dimensions, at various levels of the taxonomic hierarchy, rather than diagnostic categories (Kotov et al., 2020). To use such crude categories, which do not adequately reflect the clinical heterogeneity of psychoses, for computing dimensional scores of clinical trajectories remains confusing.
4. The authors apply sequence analysis and multidimensional scaling to investigate patterns in comorbid diagnostic trajectories in schizophrenia and other (non-affective) psychotic disorders – it needs to be clarified why this was preferred to, for example, latent growth mixture models, which would more directly address the study aims and provide more readily interpretable findings.
5. Unfortunately, this paper ignores the important role of socio-environmental factors in shaping the clinical heterogeneity of schizophrenia and other psychotic disorders. At the very least, this should be acknowledged as an important limitation of this study if not addressed in the analysis.

References

- Cross-Disorder Group of the Psychiatric Genomics Consortium. Identification of risk loci with shared effects on five major psychiatric disorders: a genome-wide analysis. *Lancet*. 2013;381(9875):1371-1379.
- Kotov R, Jonas KG, Carpenter WT, Dretsch MN, Eaton NR, Forbes MK, Forbush KT, Hobbs K, Reininghaus U, Slade T, South SC, Sunderland M, Waszczuk MA, Widiger TA, Wright AGC, Zald DH, Krueger RF, Watson D, Hi TOPUW. Validity and utility of Hierarchical Taxonomy of Psychopathology (HiTOP): I. Psychosis superspectrum. *World Psychiatry*. 2020;19(2):151-172.

Reviewer #3 (Remarks to the Author):

This is high-quality longitudinal study on differences in trajectory of schizophrenia. This topic is of great interest in psychiatry, and this study would potentially inform nosology refinement. Main strengths of this study is the use of a birth cohort of individuals who received a diagnosis of schizophrenia later in life, and the use of a data-driven statistical approach. A main limitation is the use of clinical rather than standardised research-based diagnoses. Also, individuals in this cohort could still develop schizophrenia or other psychiatric disorders. For these reasons, results are interesting but they should be cautiously interpreted. They show firstly that psychiatric comorbidity was extremely high in schizophrenia. Second, the authors performed multidimensional scaling and clustering based on dissimilarity of psychiatric comorbidity sequences. Based on different patterns of association of these enhanced phenotypes with risk factors and clinical outcomes of schizophrenia, the authors suggest that it is possible to discern different trajectories of the disease.

Introduction

The introduction is well structured but, overall, it could be more accurate. I highlighted here a few points, as follows:

Line 72 – reference no.11 is wrong: in their paper, Tandon et al. comment the decision to eliminate schizophrenia subtypes in DSM-5, which was due to the relative infrequent diagnoses of the disorganised and catatonic subtypes compared with the paranoid subtype, and not to 'longitudinal instability' as the authors claim. A major issue of operating a clear-cut division of schizophrenia into discrete sub-categories was how to classify intermediate forms, whereas the authors seem to focus only on longitudinal instability to show the poor validity of these constructs.

Line 71-74 – I find it a bit difficult to follow the entire passage, at least in the current form. The authors discuss about subtypes of schizophrenia, but then they refer to studies on continuous symptom dimensions, e.g. they cite Fanous et al. on the association between symptom dimensions and polygenic liability to schizophrenia, as attempts 'to find subgroups based on symptomatology'. Actually, the symptom dimension approach, differently from subtyping, serves primarily to scale individuals within the spectrum according to the continuous distributions of symptoms.

Line 60 – It may be worth to make clearer that biomarkers in psychiatry have been historically excluded from classificatory principles since unknown. In the same sentence, the authors explain that the categorical and syndromic approach for psychiatry classification is based on (...) 'self-description' (?). Do the authors mean that current classification criteria are based on observed and reported psychiatric symptoms? Please clarify.

Methods and results

My main concern is that there could be a higher chance of having schizophrenia in the older individuals of this cohort. Indeed, the authors correctly mention in the discussion the possibility that some study participants might still develop schizophrenia (and those with schizophrenia might still receive other psychiatric diagnoses, and vice versa); nevertheless, it would be inaccurate to describe this study as a 36 years follow up study, as this is the case only for the older individuals. Indeed, if I am right, it seems that the youngest individuals here were 7 year old at the reference time point of the Danish Psychiatric Central Research Register consultation (2012) and 11 year old at the end of the follow-up – line 351].

Another concern is related to the diagnosis reliability. Since this study uses clinician-based rather than research-based diagnoses, could it be that part of the longitudinal diagnostic instability was due to different diagnostic practices of psychiatrists? Furthermore, how changing criteria from ICD-8 to ICD-10 has impacted on changes of diagnoses of schizophrenia and other psychiatric disorders in part of this cohort over time?

Finally, it is not easy to understand what the five clusters, that are in turn based on three dimensions of comorbidities, would represent at an individual level. What are the practical advantages of this final step (clustering)? For the purposes of this study, would it be enough examining the association between risk factors and the scores at the principal MDS dimensions, rather than also clustering individuals?

Discussion

A data-driven method was used for main analyses, nevertheless it would be good to further develop discussion on some of the reported results, at least to show coherence with a theoretical framework and to strengthen the conclusions. For example, the authors explain that 'having a disease course characterised by multiple cooccurring diagnoses may be influenced by parental history of mental disorders as well as hospital treated infections, whereas comorbid substance abuse in particular may be influenced by birth weight and length.' Why these specific factors should be associated with the reported particular disease pattern?

Point-by-point response to reviewer comments

Reviewer #1:

Reviewer Comment 1: The main strength of this study is the sample, which is large, longitudinal, and representative of the population studied. The analysis approaches are fairly standard, albeit largely exploratory, and apparently sound. Support is found for some of the findings in other samples, but mainly as directional trends rather than significant findings.

Author's reply: Thank you for the comments regarding the strength of these unique data resources. Although the analytic components of our study (SA, PGS, register data, MDS, etc.) were not invented specifically for this work, we do feel this particular integration with sequential analysis constitutes a novel study design when taken as a whole. We do note the presence of a number of highly significant associations with the MDS scores after a Bonferroni correction, and hence we do not characterize the main results of the paper as suggestive trends only.

Reviewer Comment 2: Many of the findings are consistent with past epidemiologic studies showing that schizophrenia, like other mental health diagnoses, is usually comorbid with others; and that paternal age, maternal health, prenatal growth and nutrition, and childhood infections contribute to risk for schizophrenia. The case clusterings based on patterns of comorbidity are somewhat new, but largely recapitulate long-known distinctions between affective and non-affective psychoses, earlier onset of illness, and the grave impact of substance abuse both before and after diagnosis

Author's reply: It is true that some of the case clustering recapitulates known distinctions (Buckley et al.⁶), which have previously mainly been investigated with a single exposure at a time. However, our unsupervised (exploratory) approach did not necessarily *have to* recapitulate those patterns. Doing so, we feel, lends face-validity to our overall approach and allows us to speculate a bit more strongly about some patterns which appear more novel, such as the abundance of childhood disorders, multi-comorbidity and the differential associations with schizophrenia risk factors. The partial recapitulation also validates our hypothesis that the current diagnostic system for psychiatric disorders, while highly imperfect, does contain useful information for clinical course.

Regarding past epidemiological studies linking Schizophrenia to other mental disorder diagnoses, paternal age, maternal health, prenatal conditions, and infections – it is our understanding that these prior studies focused on the variables as risk factors for *disease onset* or, put another way, discriminated cases from controls (e.g. Benros et al.⁹). In fact, in our application we purposely selected these as candidate features *because* they had previous connections to schizophrenia etiology/case-control discrimination. In contrast to these previous studies, our study assesses the ability of these risk factors to discriminate particular longitudinal profiles *among schizophrenia cases*, and so we feel this extends and adds context to previous findings. Moreover, few if any pre-existing studies have examined all of the potential relationships in the same population in one integrated analysis, as we have done here.

We acknowledge that the wording in the submitted version of the abstract mischaracterizes this novelty of our study and we have modified it to say, rather than *replicate* previous finding, we *extend* them.

Text Change: Abstract: "This extends results of previous published associations (e.g., infections) and suggests that the presentation and course of schizophrenia may relate to heterogeneity in etiological factors including family history of mental disorders."

Reviewer Comment 3: The largely negative associations with psychiatric PGS are a great disappointment but may reflect limitations inherent in the original case-control samples on which the PGS weights are based. Lacking strong PGS or other genetic findings, however, I don't think the conclusion that "new likely causal relationships" have been identified is well supported.'

Author's reply: We agree that the largely negative findings between psychiatric PGS (Polygenic Risk Scores) and schizophrenia heterogeneity could be seen as a disappointment. We see a commonly held hypothesis in the field that these PGS will be good predictors for discriminating among "case types"; however, the current literature stemming from this, is somewhat mixed (e.g., Fanous et al.¹⁴ and Ruderfer et al.²⁰ both found higher polygenic liability in schizophrenia cases with more negative/ disorganized symptoms, but for instance Kalman et al.⁴⁸ and Ruderfer et al.²⁰ found no significant association with age-of-onset in Bipolar Disorder and Wimberley et al.⁴⁷ found no difference in SCZ-PRS between treatment resistant cases and other schizophrenia cases).

A complimentary hypothesis is that genetic factors that most strongly associate with case-case differences, may not be the same as those that most strongly associated with case-control differences, from which the current PGS derives. In fact, education-associated genes have been associated with case-case difference for SCZ vs. BIP (Bansal et al.,2018) and inverse genetic correlation with ADHD and ASD (Anttila et al.¹⁸). This is why we also included non-case-control PGS such as educational attainment, depressive symptoms, etc., so that we could also explore this hypothesis. In this light, we see the strong associations with education PGS (which we then replicated) and relative lack of associations with case-control psychiatric PGS as a point of interest. The reasons for this could vary and may include genetically associated environmental risk or resilience or orthogonal neurological resilience. We now make these conclusions clearer in the Discussion section by modifying the following section:

Text Change: Discussion: "While polygenic scores for education and schizophrenia both discriminate schizophrenia cases from controls in this sample, only the education score was associated with the structure of differences in comorbidity trajectories among schizophrenia patients. Relatively few studies^{14,20,45} have pursued analyses that do not focus on the primary disorder polygenic score, a common design (e.g.^{46,47}). While intuitive, there is not a theoretical requirement to believe case-control associated variants are the only relevant predictors of heterogeneity in outcomes. In fact, while some studies found SCZ-PGS to be related to chronicity⁴⁶ and negative or disorganized symptoms,^{14,20} several well powered studies have been unable to find primary disorder related PGS to be associated with clinical heterogeneity including treatment response in schizophrenia⁴⁷ and age of onset in bipolar disorder.⁴⁸"

References:

Bansal, V., Mitjans, M., Burik, C. et al. Genome-wide association study results for educational attainment aid in identifying genetic heterogeneity of schizophrenia. Nat Commun 9, 3078 (2018).

<https://doi.org/10.1038/s41467-018-05510-z>

Reviewer #2

Reviewer Comment 1a: This study focused on schizophrenia and other non-affective psychotic disorders (ICD-10, F2) – what was the reason for ignoring affective psychotic disorders (ICD-10, F3, psychotic codings)

given there is substantial overlap in genetic and socio-environmental risk among affective and non-affective psychoses (Cross-Disorder Working Group of the PGC, 2013)?

Author's reply: We agree that investigating the etiological heterogeneity in the entire spectrum of psychotic disorders (e.g. ICD-10: F21-F29, F30.2, F31.2, F31.5, F32.3 and F33.3) is an interesting research idea. In this study, we focused on the nosologically more uniform entity 'Schizophrenia' (ICD-10 subchapter: F20) for the following reasons:

- 1) Due to the case-cohort design of the iPSYCH 2012 study, we had a population-wide sample of F20 cases, while for other groups of patients with psychotic disorders (F21-F29) we had only those sampled as part of the random population cohort (~2%). We are thus much better powered and have far more valid population inferences for F20 schizophrenia.
- 2) While heterogeneity across traditional diagnostic boundaries is expected on the basis of genetic correlations across boundaries being significantly less than 1 (Schork et al.¹⁹) and GWAS associated variants appearing to discriminate along these same boundaries (Ruderfer et al.²⁰), describing genetic or etiological heterogeneity within a more narrowly defined diagnosis, such as F20 schizophrenia, is novel.

Reviewer Comment 1b: To refer to ICD-10 F2 as diagnosis of 'schizophrenia' is, strictly speaking, wrong, as several other non-affective psychoses are grouped in this category. This is somewhat surprising to read in a paper intending to focus on clinical heterogeneity and comorbidity.

Author's reply: The study is based on the narrow definition of schizophrenia "F20" (i.e F20-F20.9), as outlined in Results (line 101) and in the Methods (line 353). We have emphasized this throughout the manuscript by calling it 'ICD-10: F20-F20.9' – which may be more consistent with prior literature.

Reviewer Comment 2: Given the large sample size of the primary naturalistic cohort, it remains unclear why this was not used to generate two random samples for cross-validation to ensure findings observed in an estimation sample are directly replicated in a (statistically independent) validation sample rather than in a non-equivalent cohort recruited in a different time frame. Please clarify.

Author's reply: The methods used in this study (Sequences Analysis, Multidimensional Scaling and clustering) all fall under the category of *unsupervised learning*. Unsupervised methods do not have a risk of overfitting the effects of external explanatory variables (in our case the schizophrenia risk factors and outcomes) since they are not included in the data reduction/learning procedure. The procedure of splitting the sample into a training and a validation set (e.g., K-fold cross-validation) is generally used in *supervised* learning methods, where parameters of interest for inference are tuned for prediction (i.e., a part of the learning procedure) creating an inherent risk of overfitting. State-of-the-field in unsupervised methods estimate the stability of the resulting model fits via resampling-based methods, such as the jackknife for multidimensional scaling and resampling-based assessment of cluster stability approaches we have included (de Leeuw 1986, Hennig 2007).

Having said this, we do in fact perform a validation of our results, in the independent second cohort (diagnosed with schizophrenia 2013-2016) held out of the primary analyses. While iPSYCH2012 was extracted from the national health registries Dec 31 2012 and included all patients with schizophrenia in the age cohort in Denmark by the end of 2012, the data was later updated with follow up until Dec 31 2016, which included 870 individuals not originally part of the schizophrenia sample had been assigned a diagnosis of schizophrenia. We saw this as an opportunity to replicate our initial search for etiological heterogeneity,

without compromising the epidemiological validity of the original entire-case population ascertainment. The fact that the replication cohort was slightly different from the primary cohort in terms of diagnosis years, as we interpret it, should imply there are extra components of variance that may only *weaken or lower* the probability that a true finding would replicate. In any case we do not believe this leads to an increase chance of Type-I error in the replication, and in fact demonstrates the reliability of our findings. Thus, we interpret our replication, which in fact gave largely similar results, as a likely underestimate of (or lower bound on) the robustness of our findings.

Text Change: Results: “Additionally, we sought replication in a smaller independent sample of patients (N=870) ascertained from other iPSYCH-cohorts²⁸ and diagnosed with schizophrenia after 2012. This cohort does not have the same population representativeness of the initial cohort due to its ascertainment which should lead to underestimates of the robustness of our initial findings. Regardless, we found sign concordance for 12 of the 13 univariate dimensional associations (Table S10) and the results were strictly significant for five (education-PGS ($p= 0.0085$), maternal age ($p= 0.0003$), maternal smoking during pregnancy ($p=0.0007$), number of hospitalizations ($p=0.0002$) and total time hospitalized ($p= 0.0004$, See Table S9). Taken together, extensive sensitivity analysis and replication suggest the associations between patient heterogeneity and etiological factors are stable and reproducible.”

Reviewer Comment 3: The diagnostic states defined remain rather crude, as each state corresponds to a period of one year. Clinical states of people with psychosis can vary markedly throughout the course of a year.

Author’s reply: We had the same concern as the reviewer regarding the sensitivity of our analysis to the choice of one-year diagnostic increments. We had already performed two sensitivity analyses (data not shown in the original submitted manuscript) to test the dependence of the results on this choice. We estimated the case-case dissimilarity matrix (i.e., the key piece of information on which all of the findings of the paper are built) from subject trajectories defined by time intervals of 6 months and 4 months. We found that these manipulations resulted in highly concordant patient dissimilarity estimates, with the correlation of the dissimilarities computed with 1 year and 6 month windows being 0.9991 and with 1 year and 4 month windows being 0.9987. We have now added these sensitivity analyses to the Supplementary Materials section.

Text Change: Results: “Lowering time increments to six or four months gave dissimilarities highly correlated with those obtained using one-year increments ($r_{\text{Pearson}}=0.9991$, $r_{\text{Pearson}}=0.9987$, respectively).”

We acknowledge that diagnostic codes can be seen as crude. Therefore, we believe that trajectories of diagnoses are better suited for describing the life course heterogeneity while measures that focus more directly on symptoms presenting exactly around the time of examination might be better suited for investigating symptom variation over shorter time windows. The goals of this study, however, were more about life-course heterogeneity and its relation to etiology, where we feel the crudeness of the diagnostic codes is less relevant, especially given the unique population-complete sample with no concomitant ascertainment bias. A study with regular evaluations of symptom levels in a longitudinal cohort with long-term follow up would be an interesting study and funding such studies should be a priority. But to the best of our knowledge this currently does not exist at scale and on a population-unbiased level.

We have added a paragraph about these limitations to the discussion:

Text Change: Discussion: “Another potential limitation is that we cluster patients according to changes of diagnostic states in one year windows, rather than using finer grained symptom scales in shorter time intervals (e.g., Kotov et al.¹⁵ or Dwyer et al.¹⁶). We acknowledge that diagnoses may not capture the full nuance of a change in a patient’s presentation over time and thus think our results speak to how etiological differences may contribute to longer term, life-course presentations.”

Reviewer Comment 4: There is also substantial evidence that the heterogeneity of schizophrenia and other psychoses is accounted for by symptom dimensions, at various levels of the taxonomic hierarchy, rather than diagnostic categories (Kotov et al., 2020). To use such crude categories, which do not adequately reflect the clinical heterogeneity of psychoses, for computing dimensional scores of clinical trajectories remains confusing.

Author’s reply: As mentioned above, we agree that symptom scale heterogeneity is important and interesting; however, it can also be seen as a somewhat orthogonal or complementary view of heterogeneity in schizophrenia. So, while interesting and important, we do not agree that our lack of symptom level data invalidates the results. We discuss this more explicitly now in the manuscript.

Text Change: Discussion: “Our study could be viewed as somewhat limited by the use of health register diagnoses as opposed to standardized research-based diagnoses. Registers provide complete coverage of public hospital in- and outpatient diagnoses, but treatments by private psychiatrists or general practitioners are not recorded.⁵¹ This limitation may be mitigated by the notion that more severe cases, such as those diagnosed with schizophrenia, tend to be treated within the hospital system.⁵¹ Numerous prior studies have shown the diagnoses to be reliable, in general.⁵¹⁻⁵⁵ This may reflect that registered diagnoses are consensus assignments made at the end of hospitalization by the college of specialized psychiatrists and based on the full clinical course and set of examinations, treatments and outcomes. Although we see evidence for the change in diagnostic system from ICD8 to ICD10 occurring in 1994 in the rates of specific comorbidities among patients, our specific results appear robust to this effect. Future extension of this work should still be mindful of potential complications.

Another potential limitation is that we cluster patients according to changes of diagnostic states in one year windows, rather than using finer grained symptom scales in shorter time intervals (e.g., Kotov et al.¹⁵ or Dwyer et al.¹⁶). We acknowledge that diagnoses may not capture the full nuance of a change in a patient’s presentation over time and thus think our results speak to how etiological differences may contribute to longer term, life-course presentations. We feel this is still important for conceptualizing how to manage long term care and is somewhat complementary to investigations of symptom scales on short timelines. Further, the potential limits of cruder diagnostic changes may be balanced by the strengths of our unique data resource - long follow up, population wide, birth cohort – that are not currently mirrored in cohorts with deeper symptom-scale phenotyping. Developing such cohorts would be highly informative and complement our work here.”

Reviewer Comment 5: The authors apply sequence analysis and multidimensional scaling to investigate patterns in comorbid diagnostic trajectories in schizophrenia and other (non-affective) psychotic disorders – it needs to be clarified why this was preferred to, for example, latent growth mixture models, which would more directly address the study aims and provide more readily interpretable findings.

Author’s reply: We thank the reviewer for his/her suggestion. Our data present difficulties in terms of standard longitudinal models for three primary reasons: first the data are not just longitudinal, but outcomes are also multivariate in the sense that each ICD F Chapter is included (nine total chapters). Multivariate

longitudinal analyses using some variant of growth curve modeling (e.g., latent growth mixture models) would be possible but computationally difficult with nine trajectories for each subject, much less distilling the resulting fits into interpretable parameters; 2) there is no guarantee for linearity, in fact it is highly unlikely over the lifespan, so latent growth models would have to encompass non-parametric curve estimation, i.e. Functional Data Analysis. Third, events are right censored so standard trajectory models would be biased without accounting for this. Fourth, and most importantly, the outcomes of this study are not dimensional, but rather categorical states. In fact, considering the number of F chapters and their possible combinations, the outcome space consists of hundreds of possible categorical states. We are not aware of any existing latent growth modeling approaches that are appropriate in this context. On the other hand, Sequence Analysis, followed by MDS, which we have implemented here, was devised precisely for this type of application.

Text change: Methods: "Sequence analysis is ideally suited to the complex nature of these data, in that it can measure dissimilarities among trajectories constructed with multiple (8) categorical outcomes."²² "

Reviewer Comment 6: Unfortunately, this paper ignores the important role of socio-environmental factors in shaping the clinical heterogeneity of schizophrenia and other psychotic disorders. At the very least, this should be acknowledged as an important limitation of this study if not addressed in the analysis.

Author's reply: While other measures of social environment (e.g. household income or parental education or marital status) might impact the clinical heterogeneity in schizophrenia, these can also be potential mediators of genetic and environmental risk factors, and therefore it is not clear that including these as covariates would always be the right thing as they can be on the causal pathway. Further, our investigations were focused on outcomes that are stable or change over years and could be predicted by more life-stable "constitutional factors". We discuss this in greater detail in the updated manuscript:

Text change: Discussion: All three dimensions showed some associations with variables that are known to be correlated with socio-economic factors, such as income level and parental education, which may affect the home-environment (e.g., parental age, maternal smoking during pregnancy, education-PGS, parental psychiatric disorders). However, the direction of causality is difficult to infer.³⁹⁻⁴¹ While socio-economic factors and the home environment may directly influence disorder course (e.g. Wimberley et al.⁴²), any genetic factors associated with specific, life-trajectories of schizophrenia could alternatively affect the home environment. This second scenario could occur if a more debilitating form of the disorder leads to more home disruption or if parents, who by way of Mendelian inheritance would carry these same genes, expressed any behavioral changes that affect the home environment. This kind of gene-environment correlation across generations can be tested by studying non-transmitted alleles as has been done for educational attainment⁴³. We should note that in this study, relationships between environmental risk factors and family history of psychiatric disorders are not completely confounded, implying that environmental factors are likely to make independent contributions to disorder course. The notion of independent effects of socio-economic factors and parental history of psychiatric disorders has found previous support in contributing to risk for schizophrenia,⁴¹ and provide an interesting motivation for future gene by environment or causal inference studies of heterogeneity in outcomes and disorder course for psychiatric patients. Given the current weight of evidence, it is more likely that the home environment modifies disease course, rather than the other way around, although more work is needed, especially given the emerging perspectives of cross-generation generational genetic effects confounding intuitive causal relationships. "

Having said this, we acknowledge SES is important. We were mostly interested in biological heterogeneity. In the discussion we do mention the possible role of SES. We unfortunately did not have access to this data from the Danish national registers, which is considered sensitive information and not available for analyses without special permissions. We have added to the Discussion the limitations of not having these more direct measures of social environment available.

Text Change: Discussion: “... but we feel that more far reaching searches, including other measure of social environment (e.g. . household income or parental education or marital status), copy number and structure variants, broader collections of clinical factors, and PRS from multiple traits and diseases are needed to more fully characterize the nature of disorder course heterogeneity in schizophrenia.”

Reviewer #3 (Remarks to the Author):

This is high-quality longitudinal study on differences in trajectory of schizophrenia. This topic is of great interest in psychiatry, and this study would potentially inform nosology refinement. Main strengths of this study is the use of a birth cohort of individuals who received a diagnosis of schizophrenia later in life, and the use of a data-driven statistical approach.

Reviewer Comment 1: A main limitation is the use of clinical rather than standardized research-based diagnoses. Also, individuals in this cohort could still develop schizophrenia or other psychiatric disorders. For these reasons, results are interesting but they should be cautiously interpreted.

Author's reply: We agree that standardized research-based diagnoses are superior to clinically assigned diagnoses from national registers. However, it is inconceivable to perform a longitudinal study with up to 36-year follow-up on the entire set of individuals assigned a schizophrenia diagnosis in an entire country based on standardized diagnoses. Such a dataset does not currently exist to the best of our knowledge. Furthermore, the reliability and temporal stability of clinical diagnoses in Denmark have been extensively validated (Mors et al.⁵¹, Löffler et al.⁵², Jakobsen et al.⁵³, Bock et al.⁵⁴, Mohr-Jensen et al.⁵⁵). We also acknowledge that our results only hold for diagnoses of schizophrenia before the age of 36. We have added text speaking to the potential limits of register diagnoses in the strengths and limitations section of the discussion.

Text Changes: Discussion “Our study could be viewed as somewhat limited by the use of health register diagnoses as opposed to standardized research-based diagnoses. Registers provide complete coverage of public hospital diagnoses, but treatments by private psychiatrists or general practitioners are not recorded⁵¹. This limitation may be mitigated by the notion that more severe cases, such as those diagnosed with schizophrenia, tend to be treated within the hospital system⁵¹. While we cannot directly validate the register diagnoses used in this study, numerous previous studies have shown the diagnoses to be reliable, in general.⁵¹⁻⁵⁵ This may reflect that registered diagnoses are consensus assignments made at the end of hospitalization by the college of specialized psychiatrists and based on the full clinical course and set of examinations, treatments and outcomes. Although we see evidence for the change in diagnostic system from ICD8 to ICD10 occurring in 1994 in the rates of specific comorbidities among patients, our specific results appear robust to this effect. Future extension of this work should still be mindful of potential complications.

Another potential limitation is that we cluster patients according to changes of diagnostic states in one year windows, rather than using finer grained symptom scales in shorter time intervals (e.g., Kotov et al.¹⁵ or Dwyer et al.¹⁶). We acknowledge that diagnoses may not capture the full nuance of a change in a patient's presentation over time and thus think our results speak to how etiological differences may contribute to longer term, life-course presentations. We feel this is still important for conceptualizing how to manage long term care and is somewhat complementary to investigations of symptom scales on short timelines. Further, the potential limits of cruder diagnostic changes may be balanced by the strengths of our unique data resource - long follow up, population wide, birth cohort – that are not currently mirrored in cohorts with deeper symptom-scale phenotyping. Developing such cohorts would be highly informative and complement our work here.”

Introduction

Reviewer Comment 2: The introduction is well structured but, overall, it could be more accurate. I highlighted here a few points, as follows:

Line 72 – reference no.11 is wrong: in their paper, Tandon et al. comment the decision to eliminate schizophrenia subtypes in DSM-5, which was due to the relative infrequent diagnoses of the 8 disorganized and catatonic subtypes compared with the paranoid subtype, and not to ‘longitudinal instability’ as the authors claim. A major issue of operating a clear-cut division of schizophrenia into discrete sub-categories was how to classify intermediate forms, whereas the authors seem to focus only on longitudinal instability to show the poor validity of these constructs.

Author's reply: From our read of Tandon et al.¹¹, it appears the authors make both the argument that subtypes are not stable *and* are used infrequently as reasons for their elimination:

Tandon et al.¹¹ (page 5, 2. Paragraph): “In summary, the classic DSM-IV subtypes of schizophrenia provide a poor description of the heterogeneity of schizophrenia, have low diagnostic stability, do not exhibit distinctive patterns of treatment response or longitudinal course, and are not heritable. Except for the paranoid and undifferentiated subtypes, other subtypes are rarely diagnosed. As a result, these subtypes of schizophrenia were eliminated from DSM-5. “

We have now changed this sentence to more accurately reflect this citation and incorporate others to more broadly describe thoughts on their elimination.

Text change: Introduction: “The idea of subtypes within schizophrenia dates back more than a century³, but previous subtypes were used infrequently¹¹, had modest longitudinal stability¹², inadequately described symptom heterogeneity, and have been abandoned in DSM-5¹¹ and ICD-11.¹³”

Reviewer Comment 3:

Line 71-74 – I find it a bit difficult to follow the entire passage, at least in the current form. The authors discuss about subtypes of schizophrenia, but then they refer to studies on continuous symptom dimensions, e.g. they cite Fanous et al. on the association between symptom dimensions and polygenic liability to schizophrenia, as attempts ‘to find subgroups based on symptomatology’. Actually, the symptom dimension approach, differently from subtyping, serves primarily to scale individuals within the spectrum according to the continuous distributions of symptoms.

Author's reply: We have updated this paragraph for clarity in motivating our work.

Text change: Introduction: "The idea of subtypes within schizophrenia dates back more than a century³, but previous subtypes were used infrequently¹¹, had modest longitudinal stability¹², inadequately described symptom heterogeneity, and have been abandoned in DSM-5¹¹ and ICD-11.¹³ This reflects initiatives to describe the clinical heterogeneity in schizophrenia on symptom dimensions¹⁴ and factor analysis of symptom dimensions.¹⁵ However, these approaches have been limited by their cross-sectional design (e.g., Picardi et al.¹⁵), short follow-up (e.g., Dwyer et al.¹⁶), or retrospective data collection (e.g., Strous et al.¹⁷). Despite this, interest in describing heterogeneity in schizophrenia persists⁷ and longitudinal cohorts, with broad phenotyping, and representative ascertainment are well poised to make important contributions to this topic."

Reviewer Comment 4:

Line 60 – It may be worth to make clearer that biomarkers in psychiatry have been historically excluded from classificatory principles since unknown. In the same sentence, the authors explain that the categorical and syndromic approach for psychiatry classification is based on (...) 'self-description' (?). Do the authors mean that current classification criteria are based on observed and reported psychiatric symptoms? Please clarify.

Author's reply: Yes, that is correct, we have edited this sentence for clarity.

Text change: Introduction: "Psychiatric disorders have been classified for close to a century using a categorical and syndromic approach based on subjectively observed and reported psychiatric symptoms, rather than objective biomarkers, which limits their specificity and utility for guiding interventions.^{1,2}"

Methods and results

Reviewer Comment 5:

My main concern is that there could be a higher chance of having schizophrenia in the older individuals of this cohort. Indeed, the authors correctly mention in the discussion the possibility that some study participants might still develop schizophrenia (and those with schizophrenia might still receive other psychiatric diagnoses, and vice versa); nevertheless, it would be inaccurate to describe this study as a 36 years follow up study, as this is the case only for the older individuals. Indeed, if I am right, it seems that the youngest individuals here were 7 year old at the reference time point of the Danish Psychiatric Central Research Register consultation (2012) and 11 year old at the end of the follow-up – line 351].

Author's reply: We agree; the concern for truncation and right censoring is sound and relates to all observational and epidemiological studies. Importantly, numerous analytical approaches have been developed to analyze a cohort of differently aged individuals observed over time in order counteract the drawbacks correctly described by the reviewer. Here, we probabilistically impute the mean future diagnoses (up to age 36) based on the full sample using a Markov model.

This was a case-only study, so all individuals in the study have schizophrenia. When we talk about the possibility that some individuals may still go on to develop schizophrenia, we are referring to the whole birth cohort (all individuals born in Denmark 1981-2002). As we included all individuals born in Denmark May 1981 to December 2002 diagnosed with schizophrenia before the end of 2012 and with follow-up until the end of 2016 it is true that some individuals had follow-up of less than 36 years (range 16 to 36 years), however since schizophrenia is usually diagnosed in early adulthood only 1.1% of cases had less than 20

years of follow-up (Table 1). Thus, while unlikely, it's possible that the results might substantially change once the entire cohort ages to 36 years and over. Having said this, in our sensitivity analyses, we consider only those subjects who are 25 years of age at the end of follow-up and the results of the full sample are largely replicated. (8th paragraph in Results section)

We have edited the Method section to make the follow-up clearer and to emphasize the epidemiological nature and corresponding challenges of the study and have addressed these concerns in the Discussion.

Text change: Methods: "Using the unique Danish personal registration number (CPR), data on all psychiatric diagnoses assigned before December 31, 2016 were obtained from The Danish National Patient Register (DNPR)⁵⁶ and the PCRR.⁵¹"

Text change: Discussion: "...the use of complete longitudinal records of all hospital-assigned diagnoses and admissions over 16-36 years of follow up..."

Reviewer Comment 6: Another concern is related to the diagnosis reliability. Since this study uses clinician-based rather than research-based diagnoses, could it be that part of the longitudinal diagnostic instability was due to different diagnostic practices of psychiatrists? Furthermore, how changing criteria from ICD-8 to ICD-10 has impacted on changes of diagnoses of schizophrenia and other psychiatric disorders in part of this cohort over time?

Author's reply: The reviewer raises important points and we agree that the use of clinical diagnoses from register is an important limitation, as we discussed in response to point 1, from this reviewer.

We can more fully speak to the final concern regarding diagnostic changes. Since all individuals included in the study were born after May 1st, 1981 none were diagnosed with schizophrenia using ICD-8 criteria (before 1994). However, the oldest patients in the cohort were 13 years old when the ICD-10 was introduced in Denmark, so although a small window for a small segment of patients, some cohort effects, especially in comorbid diagnoses may persist. To specifically address this possibility, we have added a supplementary note (Supplementary Materials Section 7) and Supplementary Figures S10 and S11 which show rates of comorbidity stratified by birth-year. While we observed some trends, our sensitivity analysis in supplementary text, suggests this does not drive our results. This is because our sensitivity analyses were based exclusively on the subset of the cohort that was born before 1991, which should minimize the impact of cohort effects.

We have pointed to these data and analyses in the Results section:

Text changes: Results: "Further, diagnostic practice may change over time which could result in birth cohort effects. We calculated the cumulative incidence of each comorbidity stratified by birth year. We observed the cumulative incidence of comorbid mood disorders, autism spectrum disorders and childhood disorders increase with birth year while comorbid personality disorders decrease (Supplementary Material Section 7, Supplementary Figure S11). Our sensitive analyses suggest, however, that this cannot account for our results (Supplementary Material Section 6-7)."

and address them in the Discussion:

Text changes: Discussion: "Although we see evidence for the change in diagnostic system from ICD8 to ICD10 occurring in 1994 in the rates of specific comorbidities among patients, our specific results appear robust to this effect. Future extension of this work should still be mindful of potential complications."

Reviewer Comment 7: Finally, it is not easy to understand what the five clusters, that are in turn based on three dimensions of comorbidities, would represent at an individual level. What are the practical advantages of this final step (clustering)? For the purposes of this study, would it be enough examining the association between risk factors and the scores at the principal MDS dimensions, rather than also clustering individuals?

Author's reply: We agree with the reviewer that from an analytic perspective, it would probably be sufficient to only present the MDS associations, as these should be more sensitive displays of the same underlying mathematical concepts. In our revision and early presentation of this work, however, we often received exactly the opposite feedback! That many people felt the MDS components were hard to intuit and clusters would more clearly demonstrate the concept that patterns in patient trajectories are reflected in etiological factors. This may be because, even in medical fields where patient classification appears continuous in nature, it is common to use categorical labels (e.g. mild, moderate and severe hypertension). A secondary advantage of the clusters is that they provide some extra information in the case that the dimensions have nonuniform distributions, but the coherence of the two analyses do not suggest that to be the case.

We more explicitly state this semi-redundancy in the cluster and MDS analysis in the text, but feel it is important to retain both for the sake of presentation.

Text changes: Results: "A conceptually similar and for some more intuitive presentation of latent, multivariate dimensions capturing heterogeneity among patients is to define concrete groups that reflect a large portion of the variability in these abstract dimensions. ... The trajectory dissimilarities were thus represented well by predominantly stable and clinically interpretable groups of patients, providing a complementary representation of patient heterogeneity."

As for the clusters relating to the individual, in figure 4B, which was not directly referenced prior, we provide the modal state sequence of the patients in each cluster, which shows the diagnoses that the plurality of patients in each cluster had at each year of follow-up. This is an attempt to translate the clusters to a prototypic trajectories at the individual level. We have now highlighted this in the main text by adding the following sentence to the Results section:

Text changes: Results: "Figure 4B shows the diagnosis that the plurality of patients in each cluster had at each year of follow-up."

Discussion

Reviewer Comment 8: A data-driven method was used for main analyses, nevertheless it would be good to further develop discussion on some of the reported results, at least to show coherence with a theoretical framework and to strengthen the conclusions. For example, the authors explain that 'having a disease course characterized by multiple cooccurring diagnoses may be influenced by parental history of mental disorders as well as hospital treated infections, whereas comorbid substance abuse in particular may be influenced by birth weight and length.' Why these specific factors should be associated with the reported particular disease pattern?

Author's reply: We agree with the reviewer that some ties into a broader theoretical framework were lacking. We expanded our discussion to relate the different dimensions to prior work on case-control discriminating risk factors, neurodevelopment theories of schizophrenia, and environmental modifiers of severity. This is reflected in the text changes below:

Text Changes: Discussion: "Our work may add additional context to well-described case-control discriminating risk²⁹. For example, dimension 1, reflecting a disorder-course characterized by multiple co-occurring psychiatric diagnoses, was particularly associated with parental history of mental disorders, lower parental age at birth, maternal smoking during pregnancy, birth complications, a history with a higher load of hospital treated infections, and a more severe disorder course as captured by both number of hospitalizations and total time hospitalized for their psychiatric illness. There has been much work previously on infections being a risk factor for schizophrenia,³⁰ which have previously been shown to increase the risk of schizophrenia in a dose-response relationship with the number of severe infections,³¹ that did not appear to be confounded by the genetic risk for schizophrenia.³² Our work suggests that beyond a diagnosis, infections may predispose to a disorder course. Along these same lines, we also find associations for risk factors that have more mixed support in the current literature, such as parental age.^{33,34} Our work may add clarity to this, suggesting the effects of risk factors are not uniform throughout the patient population and may only associate with certain segments of the patient population. Other associations did not seem supported by previous literature e.g. the finding that family history of mental disorder is associated with a disease course with multiple psychiatric comorbidities.

This work is also relevant to the position of schizophrenia as a neurodevelopmental disorder.³⁵ The second dimension, which was in particular capturing patient trajectories denoted by increased prevalence of comorbid childhood psychiatric diagnoses could reflect predominance of an earlier developmental pathway that presents in childhood, possibly overlapping with what has been described as premorbid schizophrenia symptoms³⁵. This dimension was associated with a maternal schizophrenia diagnosis. Notably, the pioneering work by Fish et al.³⁶ described the developmental pandysmaturation syndrome among newborns of mothers with schizophrenia. However, this association could also have other explanations, e.g. mother's diagnosed with schizophrenia have more frequent contact with hospitals and may receive additional monitoring from the healthcare sector such that childhood symptoms receive early medical attention. This dimension was somewhat surprisingly associated with a less severe disease course after the schizophrenia diagnosis as measured by number of admissions. This could be due to the fact that this dimension was associated with less exposure to other known risk factors. Both dimension 1 and 2 could co-adhere with subtypes of schizophrenia with a more neurodevelopmental components, where particularly dimension 1 has a higher levels of early life risk factors for schizophrenia, which could indicate that neurodevelopmental pathology plays a particularly important role for these dimensions.

Dimension 3 was primarily driven by comorbid substance use disorders. This was associated with a more severe disorder course as measured by the number of hospitalizations and also more parental psychiatric diagnosis, increased birth complications, younger parental age at patient birth, and lower PGS for educational attainment. When taken together, these risk factors may suppose a less resilient or robust environment for the patient, in that birth complications, parental age, and education level are correlated with socio-economic status and access to support³⁷. Comorbid substance use may further propel one down this more severe, pro-longed course, limiting a patient's ability to obtain and maintain treatment³⁸. Thus, this dimension could represent the presentation of patients who express schizophrenia within the context of

an unsupportive or challenging environment, a part of which may stem from etiological factors (e.g., lower education PGS and propensity for substance use).

All three dimensions showed some associations with variables that are known to be correlated with socio-economic factors, such as income level and parental education, which may affect the home-environment (e.g., parental age, maternal smoking during pregnancy, education-PGS, parental psychiatric disorders). However, the direction of causality is difficult to infer.³⁹⁻⁴¹ While socio-economic factors and the home environment may directly influence disorder course (e.g. Wimberley et al.⁴²), any genetic factors associated with specific, life-trajectories of schizophrenia could alternatively affect the home environment. This second scenario could occur if a more debilitating form of the disorder leads to more home disruption or if parents, who by way of Mendelian inheritance would carry these same genes, expressed any behavioral changes that affect the home environment. This kind of gene-environment correlation across generations can be tested by studying non-transmitted alleles as has been done for educational attainment⁴³. We should note that in this study, relationships between environmental risk factors and family history of psychiatric disorders are not completely confounded, implying that environmental factors are likely to make independent contributions to disorder course. The notion of independent effects of socio-economic factors and parental history of psychiatric disorders has found previous support in contributing to risk for schizophrenia,⁴¹ and provide an interesting motivation for future gene by environment or causal inference studies of heterogeneity in outcomes and disorder course for psychiatric patients. Given the current weight of evidence, it is more likely that the home environment modifies disease course, rather than the other way around, although more work is needed, especially given the emerging perspectives of cross-generation generational genetic effects confounding intuitive causal relationships.”

REVIEWER COMMENTS

Reviewer #1 (Remarks to the Author):

The authors have carefully and thoroughly responded to the reviewers' concerns.

Reviewer #2 (Remarks to the Author):

The authors have adequately addressed only a small number of my comments. They either simply disagree or reiterate/set out in more detail the case they have made in the original version of the manuscript. I have rarely come across a revision like this and would be more than happy to look at a second revision, but this would need to at least start to address most concerns I have raised.

Given that the focus is on narrow schizophrenia (F20-F20.9) and not all non-affective psychoses (or even non-affective and affective psychoses), the issue of heterogeneity is of limited relevance.

Reviewer #3 (Remarks to the Author):

The authors well addressed all my concerns. I do not have further comments.

Response to Reviewers

While we are pleased that Reviewer #1 and #3 found our revised manuscript satisfactory, we consider the comments of Reviewer #2 unjustified and disqualifying. The very nature of the original evaluation, that essentially asked that we collect another data set and conduct another study that align with the reviewer's interests in dimensional psychopathology, makes it challenging to respond constructively and succinctly. Two lines of critique seem to predominate in this reviewer's second set of comments; namely that we only address a minority of the original concerns and that heterogeneity in schizophrenia (F20) is without interest.

First, we very explicitly addressed the critique of our analytical approach that we should have used (a) a split-sample design to create a discovery and a replication sample, and (b) latent growth mixture models. In both instances, we clearly explained how these ideas are statistically uninformed. While we see no reason to repeat our arguments, we will be more than happy to discuss these issues with a statistically qualified expert while pointing out that one of the senior authors on this study (WKT) is himself a Professor of Biostatistics with training, expertise, and almost twenty years of experience in methodological development and application of longitudinal data analyses, including cross-validation methods and the development and application of latent growth mixture models in the contexts in which these approaches are actually useful, and who is Director of Biostatistics for two longitudinal NIH-funded US-national consortia studying neuropsychiatric outcomes: the Adolescent Brain Cognitive Development Study (<https://abcdstudy.org/>) and the National Consortium on Alcohol & Neurodevelopment in Adolescence (<http://www.ncanda.org/>).

Second, while possibly having been less explicit in our first response, we strongly disagree that "... focus is on narrow schizophrenia (F20-F20.9) and not all non-affective psychoses (or even non-affective and affective psychoses), the issue of heterogeneity is of limited relevance." In fact, we find this astonishing statement to be completely unrooted in clinical psychiatry. The diagnosis of schizophrenia (F20) is the major and paradigmatic class of non-affective, psychotic disorders in the ICD-classification systems guiding diagnostics and clinical intervention. Adding to the relevance of F20-focused studies is that heterogeneity in schizophrenia (F20) is a genuine clinical concern and an enormous challenge.

The statement is also scientifically astounding, as it dismisses decades of research in schizophrenia (F20) that has resulted in countless, high-impact publications and provided unprecedented insight into the genetic, clinical, socio-economic and -demographic as well as environmental aetiology of this diagnostic category. In fact, senior authors on this manuscript (TW, DG) have contributed massively to provide this insight. The study of clinically relevant heterogeneity within this diagnostic class is self-evidently of interest well beyond the field of psychiatry possibly inspiring analogous analytical approaches in similarly complex and highly heterogeneous disorders, for example type-2 diabetes, inflammatory bowel disease, and hypertension.

Considering the possible interest of Reviewer #2 in dimensional psychopathology, we want to emphasize that while attempts to identify and characterize such non-diagnostic traits shared across diagnostic categories is of obvious interest in psychiatry; it makes little sense to study diagnostic differences across diagnostic categories as the latter by-construction are different; i.e. such a study design will create an a priori dichotomy in the data set. Contrary, as exemplified by our analyses, the study of diagnostic differences within a single diagnostic class may inform on clinically relevant differences and their etiological underpinnings, as in fact suggested by Sir Michael Owen in his 2014-commentary to RDoC (cite; cited in the Introduction of our manuscript).

We therefore stand by our revised manuscript.

REVIEWER COMMENTS

Reviewer #4 (Remarks to the Author):

Based on reading only the abstract first, the study sounds very exciting and interesting. Also some questions arise:

1) Why only people with eventually diagnosed schizophrenia were included? How about all those people, who have had potentially similar comorbid patterns, but the "final" diagnosis of schizophrenia has never (or not yet) been set?

2) What is the role of psychiatric health care system and clinical diagnoses in co-occurring diagnoses, i.e. are the observed diagnoses reflections of several separate real mental disorders or just reflections of the difficulty to make a "correct" diagnosis or e.g. the tendency to avoid giving schizophrenia diagnosis to children? How well substance (ab)use can be extracted from the register data – are only the most extreme or otherwise selected cases captured?

3) Cluster analysis is very data-driven method. Are the results really stable as stated and do them generalize also to other data sets possible from other countries?

INTRODUCTION

Introduction seems to give partial answer to the question 2. Or maybe not the answer, but the briefly reviewed earlier research on heterogeneity in schizophrenia and justification of the research problem for the current study are well done. What I still miss is the explanation of pragmatic effect of real-world diagnoses in contrast to theoretical variation in schizophrenia.

METHODS

In the methods section, it is told that people with schizophrenia until the end of 2012 were included. Was there any particular reason for not including more recent years? I see that later in the replication sample paragraph you tell that those people were used as a replication sample. That is fine, but could you please justify why the newest cohort with partially different data availability is the best choice for replication sample, e.g. why not to keep 10% from each year since the beginning as a replication sample? Also why the definition/reference to supplementary material of diagnoses used is given here at the replication sample description – should it be already in the description of the primary sample?

When the switch from ICD-8 to ICD-10 happened? You mention that ICD-8 was used before 1994 and ICD-10 after 1994, so does that mean that both were used during 1994? Do you think it is worth of mentioning that change of disease classification did not change the definition of schizophrenia? How about the classification of other mental disorders – is it possible that changes from diagnoses given in childhood (more with ICD-8) therefore differ from the later diagnoses classified with ICD-10? I found discussion of these issues from the supplementary material chapter 7, please refer to it in the main text.

Were psychiatric outpatient and inpatient data (both) completely available since the 1981? Actually there was an answer to that in the next paragraph and outpatient visits were available since 1995. Do you think that there is a difference/selection in your patient population due the fact that only psychiatric inpatient visits were available until 1995? There was some discussion in the supplementary material chapter 7, please refer to it.

In the linkage of family histories, which years were included? Were there a lot of missing links to fathers in CPR? Do you think that maternal or paternal mental disorders were completely captured due to the restrictions of PCRR data availability?

In the survival analysis paragraph, could you please clarify/confirm that follow-up was from birth until the diagnosis of schizophrenia. So no censoring due to deaths, migration or end of follow-up were present?! Why to use survival analysis in such a case? Or do you mean that you had eight models, i.e. one for each comorbidity category and follow-up was from birth to that particular comorbidity occurrence (or censoring due to death, migration or the end of follow-up) and if that

comorbidity was not the first one, the first one was used as a time-dependent covariate? If there is more thorough description of the analysis in supplementary material, please refer to it here. I have no doubts that the analysis would be inadequate, but the description is so brief that it is difficult to follow what was actually done.

In the description of sequence analysis, there appears to be some inconsistencies in relation to supplementary material: Observation lengths: 16-36 years vs. 16-35 years and number of states (247 vs 193). Please clarify.

Please describe how state for each year was defined, i.e. was it required that comorbid diagnoses occurred during the same year (i.e. person had to have admissions with those diagnoses at the considered year) or were diagnoses cumulative so that after the first diagnosis that diagnosis remained for all subsequent years?

The technique for imputing the missing states is well described and providing the software to do that is a definitively positive thing. However, it seems that most sequences needed imputation and partly more than half of the whole sequence was imputed. In general, the predictions to near future tend to be more accurate than the ones for the distant future. What do you think, is the case similar here, i.e. that the accuracy of the imputation is getting "worse" for longer imputation periods? Is there anything you could do about it? You describe some sensitivity analyses in supplementary material chapter 6, but no results of these analyses or reference to them is presented there?! There appears to be some results in supplementary table 11.

Nice analyses for the stability of the MDS.

In the description of MANCOVA analyses, please specify how many dimensions of MDS were used (all 13 or the first 3 or something else)? In the multinomial logistic regression, how many clusters were used (was it five)?

In "association with outcome" paragraph, were your dependent variables ("the disease trajectories") the MDS dimensions (and how many) again? If so, please write it out.

DATA AND CODE AVAILABILITY

Excellent shiny app for further results and detailed R code for most of the analyses provided.

RESULTS

Here it is stated that 10864 population controls (i.e. two per cohort member) were also followed. Why this has not been mentioned in the methods, abstract or supplementary material? Why there are (minor) differences in the age and sex distributions even though the controls were matched? And please clarify: matched at which point in time (probably at the time of schizophrenia)?

In the figure 1, cumulative incidences are obviously calculated separately for each disorder as summing those up (to "combined outcomes") gives proportions over 1. Was potential censoring (due to death, migration or end of follow-up) after diagnosis of schizophrenia taken into account? Recently it has been claimed that cumulative incidence should be calculated by taking the competing risks into account to provide reasonable estimates – how would you explain why that approach has not been used here (at least for the part after schizophrenia/matching date)?

Typo in line 125: $p < 0.000$ -> $p < 0.0001$

Secondary comorbidity analyses in Figure 2. Description of the Cox models was somewhat unclear (see comments in the methods section) and also the R-code for these analyses is a bit difficult to read without testing it with some data. As mstate-package was used, were you taking competing risks into account in these analyses? Please add a bit more specific description to the methods section.

Line 141: You refer to Figure 1 while describing psychiatric trajectories. Do you mean Figure 3? What are the "Online Methods" representing here – supplementary material or R-code? If supplementary material, please refer to it so that the corresponding section can be identified from

the material.

Please specify what correlations represent while you lower time increments as reported in lines 147-148. I understand that you have dissimilarity-matrix between trajectories, but hasn't that matrix different dimensions if you change time increments so how to calculate correlation?

In the results section (line 150) it is stated that logistic regression analyses were conducted, which sounds clear for diagnoses, but in the analyses for total count of diagnoses or the age at first diagnosis you probably have utilized other than logistic regression?

The paragraph describing analyses for post hoc analyses giving interpretations to MDS dimensions is somewhat complicated to read as different approaches or scales have been reported for different dimensions. Those seem to be fine, but the question is how those interpretations were found and selected to be the best ones?

In Figure 4B, what does darker grey in cluster 1 represent (missing from the legend)?

As the clustering results in relevantly stable and interpretative groups of patients, would it be possible to give certain fixed definitions for these groups so that the similar division to groups could be done based on those rules instead of using data-driven clustering? That would allow much better repeatability of the analysis in other contexts just by assuming that groups are somehow reasonable from clinical point of view while repeating whole data-driven approach will certainly result in different groupings masking the possibility for direct comparisons.

Is there a need to report all F and P values for MANCOVA and ANCOVA analyses if those are in any case reported in Table 2. In addition, I would not in this case use scientific notation for P-values in text or (supplemental) tables. I would also prefer reporting confidence intervals at least with OR:s and HR:s.

Sensitivity analyses are comprehensive. Would it be possible to somehow indicate potential inconsistencies in the supplementary table 11? Now it is difficult to read.

DISCUSSION

I think the discussion is comprehensive and well structured. Just a few ideas that emerged while reading it:

Are there some psychiatric comorbidities that are particularly strongly associated with later schizophrenia? For example, it has been reported that half of patients with cannabis-induced psychosis have got schizophrenia diagnosis in five years.

Would it possible to include birth season as a risk factor for schizophrenia here? It has been reported that birth rate for people with schizophrenia is higher during spring or winter months.

In conclusion, the study appears to be very elegantly conducted with refreshingly different methods with novel results. There are some shortcomings in the reporting (how the manuscript has been written) and I have tried to point out issues that complicated my reading of the manuscript. I hope that those comments/suggestions help to improve the (readability of the) manuscript further, but I left it up to the authors (and editor) to decide which ones require corrections.

Dr. Reijo Sund
Professor of Register Studies,
University of Eastern Finland,
Kuopio, Finland

Reviewer #5 (Remarks to the Author):

I gather that this manuscript has been through at least one round of reviews and revisions and I will keep this in mind. However, this is my first review of the paper.

Authors present findings from a remarkable, longitudinal sample of >5400 Danish schizophrenia cases based on detailed information from Danish civil and health registries. They characterized these individuals in terms of sequences of comorbid diagnoses arising before, after, and concurrently with the focal schizophrenia diagnosis, thus creating developmental comorbidity trajectories for each case. The comorbidity trajectory data were then analyzed using MDS to identify prominent comorbidity trajectory dimensions and, further, were subjected to potentially simplifying cluster analyses to identify trajectory subgroups. Finally, authors tested the associations of both MDS dimensions and cluster-based subgroups with genetics (PGS), gestational factors, parental factors including mental health history, and certain indicators of clinical severity and course. Results highlighted the importance of (1) the number of comorbid diagnoses, (2) diagnoses of childhood disorder comorbidities vs mainly adult comorbidities, and (3) the presence of a substance abuse comorbidity. Clinical outcomes, gestational infections and other gestational issues, family history of mental illness, and education PGS, among other things, were differentially associated with dimensional differences and subgroup membership.

Heterogeneity is a challenge for clinical care and research in schizophrenia with huge implications for biological and treatment discovery. In particular, heterogeneity in developmental and clinical course in this disorder is widely recognized and fundamental. The current analyses take a novel approach to this aspect of schizophrenia heterogeneity and offer possible clues to risk, etiology and clinical outcome. To me the analytical approaches are well-matched to the data and research goals. I particularly appreciated seeing associations – for both dimensional and categorical approaches – to the family history, gestational, hospitalization, and genetic variables. My principal comments relate to discussion points.

The paragraph beginning on line 351 discusses the association of education PGS with the dimensional and categorical structure of comorbidity trajectories. One recent study (Dickinson et al., *American Journal of Psychiatry*, 2020) reported consistent findings that authors might wish to reference. Dickinson et al. found that a profile of PGS, including educational PGS, distinguished schizophrenia subgroups defined in terms of putative trajectories of cognitive development. In particular, education PGS helped distinguish a subgroup characterized by suspected early childhood impairment from other groups with different developmental trajectories, consistent with certain findings here.

Authors have marshalled and aligned large streams of longitudinal data to conduct these very interesting analyses. Limited information about outcomes, however, constrains interpretation. Authors are able to make interesting connections to hospitalization variables. Information about functional outcomes, including educational and/or vocational attainment by schizophrenia cases, would have been a valuable complement. For example, around line 291 authors indicate surprise that the subgroup with comorbid childhood disorders showed less severe hospitalization outcomes after diagnosis with schizophrenia. However, relatively lower clinical instability in this group might be accompanied by significant limitations in social and role functioning in the community (eg, Dickinson et al., 2020 – moderate symptomatology, but lower levels of education, employment and cognitive performance in early life impairment group). Perhaps these data were not readily available from public registry sources, but this seems a potentially fruitful direction for future work.

Smaller points:

Regarding line 167 'nearly half of between-patient variability in trajectory dissimilarity' – Is this accurate? Or is it nearly half of the 79% trajectory dissimilarity variance explained by the 3 MDS dimensions?

Column B in Figure 4 is a bit confusing. It would be better if there were not two shades of gray in this part of the figure. I assume that the pale gray color signifies that the modal diagnosis for that

particular follow-up period is schizophrenia. However, the meaning of pale gray color should be stated explicitly in the color key at the bottom of the column.

More generally, I found the figure captions overly terse and that they could benefit from revisions for greater clarity.

Table S 8, should there be columns for Cluster #5?

Typographical:

Should the year ranges on lines 94 and 106 be consistent?

Line 150 'assess'

Line 240 'sensitivity'?

Line 291 '. . . what have been described . . .'

Line 424 'PRS'? Or 'PGS', as used elsewhere.

Line 763 '. . . each to be assigned a cost.'?

RESPONSE TO REVIEWER COMMENTS

Reviewer #4:

Dear Dr. Reijo Sund,

We thank you for your extremely thorough and thoughtful review of our paper and its associated supplements and online resources. It led us to include a few extra simulations and extended analyses, in addition to a much improved presentation and discussion of our approach which has strengthened the paper *substantially* and provides a good foundation for future work. These ideas would not have arisen without inspiration from your comments. Note, changes from the pre-existing text are highlighted in yellow below.

Question from the reviewer:

Based on reading only the abstract first, the study sounds very exciting and interesting. Also some questions arise:

1) Why only people with eventually diagnosed schizophrenia were included? How about all those people, who have had potentially similar comorbid patterns, but the “final” diagnosis of schizophrenia has never (or not yet) been set?

Authors' response:

This poses an interesting question, but also a quite different one than addressed here. Our core question (disease trajectories) and analytical strategy (sequence alignment) are novel and address a phenomenon that has not been systematically studied in the past. Therefore, we find it important that the study design is clearly circumscribed, well-defined and anchored in decades of clinical practice as well as in scientific investigation of etiology. While the study of a single diagnostic category, such as schizophrenia, fulfills these criteria, it would not be the case for studies of broader categories of disorders. However, we are pursuing subsequent efforts intended to explore broader categorizations while keeping in mind that such studies in essence address somewhat different questions from those pursued here.

We have inserted a short statement on the rationale underlying our study question and design in the introduction and added a sentence to the limitations section.

Text changes:

In the introduction:

“To investigate if the clinical heterogeneity within clinically defined schizophrenia (ICD-10: F20.0-20.9) is linked to etiological heterogeneity, the individual patient projections onto the leading principal dimensions of trajectory dissimilarity from the MDS analysis were tested for association with known risk factors and clinical outcomes. This could help uncover biological heterogeneity and motivate more personalized clinical care to improve outcomes in schizophrenia.

In the limitations:

“While we focused on detecting and describing heterogeneity within schizophrenia, more studies are needed to uncover how this relates to other diagnostic categories, for example, broadening to a wider spectrum of disorders involving psychosis.”

Question from the reviewer:

2.1) What is the role of psychiatric health care system and clinical diagnoses in co-occurring diagnoses, i.e. are the observed diagnoses reflections of several separate real mental disorders or just reflections of the difficulty to make a “correct” diagnosis or e.g. the tendency to avoid giving schizophrenia diagnosis to children?

Authors’ response:

We agree with the reviewer that a *diagnosis* is not necessarily indicative of an underlying *disorder* (i.e., in real-world clinical practice they do not necessarily map onto each other one to one). This may be especially relevant when considering each diagnosis from all hospital contacts for a patient across a long time period and when considering phenomena around a disorder, like schizophrenia, that onsets in adulthood and represents an apex of a clinical severity hierarchy. We do believe, however, that a registered diagnosis is at least indicative of help-seeking behavior associated with an expert classified set of meaningfully disruptive symptoms (i.e., a time-stamped, valid clinical presentation). A comorbid diagnosis adjacent to schizophrenia, then, could be viewed as having arisen for many reasons, including: 1). a random expression of one underlying disorder (e.g., misdiagnosis or a chance description of general prodrome), 2). evidence of a second disorder, or 3). marking unique symptoms/presentations of variation in a single disorder (i.e., true heterogeneity).

From a data analysis perspective, then, our hypothesis was that the leading MDS dimensions and clustering approaches we adopted would be maximally sensitive to the third reason, in that the specific, transitional patterns of diagnosis before and after schizophrenia onset would be present in substantial subsets of the patients (i.e., reflect non-random transitions present in many subjects and over long time periods). On the other hand, we hypothesized that misclassification or stochastic labeling of prodromal symptoms with other diagnoses would not associate into distinct, long-term patterns of further comorbidity, thus resulting in less stable clusters which account for less overall variation in trajectories and with potentially fewer associations with (distinct) risk factors. We now discuss the lower stability of the depression only cluster in context of the possible overlap between prodromal symptoms and depression symptoms.

From a more clinical perspective, it is well described that individuals with schizophrenia often present with symptoms before onset of psychosis (i.e., prodromal symptoms) While it is hard to date the onset of prodromal symptoms in prospective studies, since patients are most often diagnosed after onset of psychotic symptoms, there seems to be some consensus that the prodromal symptoms (particularly pre-psychotic symptoms) have onset weeks to (maximum six) months before psychosis onset (Keith & Matthews, 1991). It’s not unlikely that some registered diagnoses given in the months preceding a schizophrenia diagnosis, then, will be reflective of prodromal symptoms (1) (Fusar-Poli). A canonical clinical example might be a tendency to delay the schizophrenia diagnosis in children (Okkels 2013), though the mean age of diagnosis in early onset schizophrenia has decreased over last decades (Okkels 2013), which could indicate an increased focus on early diagnosis. However, it also has previously been shown that other psychiatric diagnoses, such as childhood diagnoses, increase the risk of schizophrenia many years later and this was thought to be more indicative of either distinct disorders (2) or different presentations (3), rather than prodromal symptoms (Maibing, 2015). We feel the stability of our reported patterns and association with risk factors suggests our approach is able to return meaningful groups, even acknowledging these features of the data.

Thus, while we agree with the reviewer that interpreting patterns in register-based diagnoses requires a nuanced approach, we also see unique opportunities for indirectly tracking variation across patients in changes in symptom presentation, because of the “true to clinical practice” way in which the Nordic registers aggregate life-course diagnoses on individual patients.

We have added a paragraph to the discussion, where we reflect on one of the “null” findings of the study to present some of this nuance considering prodromal symptoms.

References:

Keith, S. J., & Matthews, S. M. (1991). The diagnosis of schizophrenia: A review of onset and duration issues. *Schizophrenia Bulletin*, 17(1), 51–67. <https://doi.org/10.1093/schbul/17.1.51>

Okkels N, Vernal DL, Jensen SO, McGrath JJ, Nielsen RE. Changes in the diagnosed incidence of early onset schizophrenia over four decades. *Acta Psychiatr Scand*. 2013 Jan;127(1):62-8. doi: 10.1111/j.1600-0447.2012.01913.x. Epub 2012 Aug 20. PMID: 22906158.

Maibing CF, Pedersen CB, Benros ME, Mortensen PB, Dalsgaard S, Nordentoft M. Risk of Schizophrenia Increases After All Child and Adolescent Psychiatric Disorders: A Nationwide Study. *Schizophr Bull*. 2015 Jul;41(4):963-70. doi: 10.1093/schbul/sbu119. Epub 2014 Sep 5. PMID: 25193974; PMCID: PMC4466169.

Text changes:

In the discussion:

“There are many reasons an individual may receive a particular diagnosis prior to schizophrenia but not all would result in diagnoses that are necessarily meaningful for considering patient groups. We found that 734 individuals had a trajectory in cluster 3 characterized by affective disorder diagnoses, mostly occurring before onset of schizophrenia. However, this cluster had low stability and only a few of the putative risk factors showed significant associations with Cluster 3, which could indicate that comorbid affective disorder diagnosis alone does not capture any specific pathology. Notably, prodromal symptoms of schizophrenia can overlap with depression symptoms (Fusar-poli, 2014). In contrast to this, clusters 1, 2, 4 and 5 were fairly stable and most of the associations with risk factors and outcomes seen in dimensional representation were also associated with clusters 1-3 when comparing to cluster 5. This indicates that features loading on these clusters (childhood disorders, multiple adult comorbidities and substance abuse) form more distinct trajectories and risk factor and outcome profiles. This could be consistent with childhood disorders and substance abuse having more symptoms clearly distinguishable from typical schizophrenia symptoms, marking more stable heterogeneity, while a prior depression diagnosis may be more epiphenomenal (e.g., relating to prodromal symptoms or particularities of clinical practice).”

Additionally, we have added a section to the limitations section emphasizing the uncertainty about psychiatric diagnoses:

“The use of psychiatric diagnoses to describe symptom presentation is subject to a number of limitations including imperfect inter-rater reliability and the reliance on imperfect classification systems. Further, our study could be viewed as somewhat limited by the use of health register diagnoses as opposed to standardized research-based diagnoses. Registers provide complete coverage of public hospital in- and outpatient diagnoses, but treatments by private psychiatrists or general practitioners are not recorded.⁵¹ This limitation may be mitigated by the notion that more severe cases, such as those diagnosed with schizophrenia, tend to be treated within the hospital system.⁵¹ Numerous prior studies have shown the overall diagnoses in the registers, including for schizophrenia, to be reliable.⁵¹⁻⁵⁵ This may reflect that registered diagnoses are consensus assignments made at the end of hospitalization by the college of specialized psychiatrists and based on the full clinical course and set of examinations, treatments and outcomes. Although we see evidence for the change in diagnostic system from ICD8 to ICD10 occurring in 1994 in the rates of specific comorbidities among patients, our specific results appear robust to this effect. Future extension of this work should still be mindful of potential complications. “

Finally, this revealed typo in the results section where we by mistake referred to the mood disorder-only cluster as cluster “4” instead of “3” and the substance abuse cluster as cluster “3” instead of “4”, which is inconsistent with Figure 4, and supplementary tables S4 and S8. This has now been corrected:

"Among the five clusters (1-5), cluster 1 (N=597) contained patients with comorbid childhood disorders (Cumulative Incidence (CI)_{F7-F9}: 100%), cluster 2 (N=1580) contained patients with multiple adult-comorbidities (CI_{F1-F6}: 99%), cluster 3 (N=729) contained patients with only mood disorder comorbidities (CI_{F3}: 100%), cluster 4 (N=734) contained patients with comorbid substance abuse (CI_{F1}: 100%), and cluster 5 (N=1792) contained patients with little comorbidity (CI_{F1-F9}: 46%) (Figure 4C). With the exception of cluster 3 (mood disorders-only) which did not separate clearly from cluster 5 (no-comorbidity) and cluster 2 (adult-disorder) (Figure S8), these clusters were stable (mean Jaccard coefficient > 0.59; Table S4)."

References:

Fusar-Poli, P., Carpenter, W. T., Woods, S. W., & McGlashan, T. H. (2014). Attenuated psychosis syndrome: Ready for DSM-5.1? *Annual Review of Clinical Psychology*, 10, 155–192. <https://doi.org/10.1146/annurev-clinpsy-032813-153645>

Question from the reviewer:

2.2) How well substance (ab)use can be extracted from the register data – are only the most extreme or otherwise selected cases captured?

Authors' response:

A previous study has found that individual admissions for substance abuse *per se* is generally underdiagnosed in Danish registers including patients admitted for schizophrenia (Hansen, 2000). However, the cumulative incidence of substance abuse comorbidity in our study (~35%) is much higher than the frequency reported in the registers in that study (13.7%), which could reflect that while substance abuse is not always diagnosed at a single (first) contact, it will often be picked up at later contacts. This may be especially true for our studied population as there are specialized departments in Denmark for schizophrenia patients with substance abuse (so-called double-diagnosis patients/departments), a standard practice for extra SUD screening with psychotic patients. We might expect, then, that the coverage in schizophrenia patients is better, but may still be somewhat under-diagnosed, and likely delayed in time.

Reference

Hansen, S., Munk-Jørgensen, P., Guldbæk, B., Solgård, T., Lauszus, K., Albrechtsen, N., Borg, L., Egander, A., Faurholdt, K., Gilberg, A., Gosden, N., Lorenzen, J., Richelsen, B., Weischer, K. and Bertelsen, A. (2000), Psychoactive substance use diagnoses among psychiatric in-patients. *Acta Psychiatrica Scandinavica*, 102: 432-438. <https://doi.org/10.1034/j.1600-0447.2000.102006432.x>

Text changes:

We have added to these points to the limitations:

"Our study could be viewed as somewhat limited by the use of health register diagnoses as opposed to standardized research-based diagnoses. Registers provide complete coverage of public hospital in- and outpatient diagnoses, but are known to have certain blind spots, for example, treatments by private psychiatrists or general practitioners are not recorded, and it has been found that substance use disorders in admissions for schizophrenia were underdiagnosed (Hansen, 2000).⁵¹ This limitation may be mitigated in our study by the notion that more severe cases, such as those diagnosed with schizophrenia will, tend to be treated within the hospital system⁵¹ and typically have multiple hospital contacts such that the register

coverage of comorbidities might be better in these patient populations. Numerous prior studies have shown the overall diagnoses in the registers, including for schizophrenia, to be reliable.⁵¹⁻⁵⁵ This may reflect that registered diagnoses are consensus assignments made at the end of hospitalization by the college of specialized psychiatrists and based on the full clinical course and set of examinations, treatments and outcomes. Although we see evidence for the change in diagnostic system from ICD8 to ICD10 occurring in 1994 in the rates of specific comorbidities among patients, our specific results appear robust to this effect. Future extension of this work should still be mindful of potential complications. “

Question from the reviewer:

3) Cluster analysis is very data-driven method. Are the results really stable as stated and do they generalize also to other data sets possible from other countries?

Authors' response:

We agree that data driven approaches like clustering can be very sensitive to the data they are conducted on, and our experience is that clustering algorithms can produce seemingly interesting grouping in data, but change dramatically when the subsets of the data are held out. To address this issue, we undertook a number of sensitivity and stability analyses which suggest our results are as robust as can be expected. In this paper we include a replication in an independent cohort and find general agreement, but we agree with the reviewer that support from a replication in an independent cohort, ideally from another country, would strengthen the results. Reviewer #5 has pointed us to a recent study from the US that finds similar trends in cross-sectional data from a clinical cohort, which could also strengthen the findings from our study. We have added these points to the discussion section that now mentions explicitly in the limitations section that results should be replicated preferably in a cohort from another country. We also provide a decision tree (detailed below in response to question 30) to enable this.

Text changes:

In the Discussion:

“This work is also relevant to the position of schizophrenia as a neurodevelopmental disorder.³⁵ The second dimension, which was in particular capturing patient trajectories denoted by increased prevalence of comorbid childhood psychiatric diagnoses could reflect predominance of an earlier developmental pathway that presents in childhood, possibly overlapping with what has been described as premorbid schizophrenia symptoms³⁵. This is in line with a recent study by Dickinson et al (Dickinson, 2020) who used a cross-sectional design and found a subgroup of schizophrenia patients with signs of a more neurodevelopmental course. This dimension was associated with a maternal schizophrenia diagnosis. Notably, the pioneering work by Fish et al.³⁶ described the developmental *paedysmaturation* syndrome among newborns of mothers with schizophrenia. However, this association could also have other explanations, e.g. mother's diagnosed with schizophrenia have more frequent contact with hospitals and may receive additional monitoring from the healthcare sector such that childhood symptoms receive early medical attention. This dimension was associated with a less severe disease course after the schizophrenia diagnosis as measured by number of admissions. Interestingly, Dickinson et al (Dickinson, 2020) also found that individuals with schizophrenia and a pre-adolescent cognitive impairment had less severe symptoms after onset, than those with more typical pre-adolescent cognitive performance that declined (adolescent disruption of cognitive development). Both dimension 1 and 2 could co-adhere with subtypes of schizophrenia with a more neurodevelopmental components, where particularly dimension 1 has a higher levels of early life risk factors for schizophrenia, which could indicate that neurodevelopmental pathology plays a particularly important role for these dimensions.”

In the limitations:

“... and our results should be replicated externally, preferably using cohorts from another country.”

Question from the reviewer:

INTRODUCTION

4) Introduction seems to give partial answer to the question 2. Or maybe not the answer, but the briefly reviewed earlier research on heterogeneity in schizophrenia and justification of the research problem for the current study are well done. What I still miss is the explanation of pragmatic effect of real-world diagnoses in contrast to theoretical variation in schizophrenia.

Authors' response:

We discuss the distinction between real-world diagnoses and theoretical variation in our answer to question 2.1, above. However, this point and similarly themed points raised by several other reviewers make it clear to us that it's important to discuss the uniqueness of register data a little more clearly in the Introduction.

Repeating our thoughts briefly, a registered hospital diagnosis can be thought of as reflecting help seeking behavior regarding a particular set of symptoms, that were judged by a trained psychiatrist to be consistent and severe enough to warrant clinical intervention. These symptoms may reflect long lasting features of an individual's personality or other traits (such as personality disorders, resilience), they may be transient states (such as depression), or they may mark unique forms of schizophrenia. These clinical diagnoses – as recorded in the registers – are *de facto* the phenomenon we wish to study, as we want to understand real-world psychiatry and feel the finding of stable, associated with risk factors, within this real-world data, is of key importance.

Text changes:

We have now added this paragraph to the introduction:

“The Danish health registers offer a unique perspective on life-course heterogeneity in the clinical presentations of schizophrenia patients. Established in 1968, the Danish registration system²²⁻²⁴ has provided nearly complete coverage of the health service usage of the complete population of Denmark for more than 50 years, including psychiatric hospital contacts²³. The Psychiatric Central Research Register (PCRR)²³ follows the population from birth in a longitudinal and prospective manner, providing a unique, time-stamped, and reliable^{23,25-28} diagnoses for an individual at each hospital contact. These data, then, more closely reflect real-world clinical practice than retrospective case-control diagnoses because they objectively catalog preceding and succeeding psychiatric contacts. Previous studies have used these powerful data to define rates of comorbid diagnoses in psychiatry⁸ and describe specific patterns of transitions among diagnoses to demonstrate difficulties classifying patients at first admission (Musliner, 2020). Danish register data have also been used to describe both prodromal states (Maibing, 2014) and premorbid traits in patients later diagnosed with schizophrenia (Maibing, 2014, Urfer-Parnas, 2009), and to record longitudinal stability in schizophrenia diagnoses (Jakobsen, 2007). A systematic, data-driven study of life course patterns of comorbid diagnoses and their relation to etiological factors has not been pursued but could contribute greatly to how we understand heterogeneity within schizophrenia.”

References:

Musliner, K. L., Krebs, M. D., Albiñana, C., Vilhjalmsson, B., Agerbo, E., Zandi, P. P., ... Østergaard, S. D. (2020). Polygenic risk and progression to bipolar or psychotic disorders among individuals diagnosed with

unipolar depression in early life. *American Journal of Psychiatry*, 177(10), 936–943. <https://doi.org/10.1176/appi.ajp.2020.19111195>

Maibing CF, Pedersen CB, Benros ME, Mortensen PB, Dalsgaard S, Nordentoft M. Risk of Schizophrenia Increases After All Child and Adolescent Psychiatric Disorders: A Nationwide Study. *Schizophr Bull*. 2015 Jul;41(4):963-70. doi: 10.1093/schbul/sbu119. Epub 2014 Sep 5. PMID: 25193974; PMCID: PMC4466169.

Urfer-Parnas, A., Lykke Mortensen, E., Sbye, D., & Parnas, J. (2010). Pre-morbid IQ in mental disorders: A Danish draft-board study of 7486 psychiatric patients. *Psychological Medicine*, 40(4), 547–556. <https://doi.org/10.1017/S0033291709990754>

Jakobsen, K. D., Hansen, T., & Werge, T. (2007). Diagnostic stability among chronic patients with functional psychoses: An epidemiological and clinical study. *BMC Psychiatry*, 7, 1–8. <https://doi.org/10.1186/1471-244X-7-41>

Question from the reviewer:

METHODS

5) In the methods section, it is told that people with schizophrenia until the end of 2012 were included. Was there any particular reason for not including more recent years? I see that later in the replication sample paragraph you tell that those people were used as a replication sample. That is fine, but could you please justify why the newest cohort with partially different data availability is the best choice for replication sample, e.g. why not to keep 10% from each year since the beginning as a replication sample?

Authors' response:

This is a very good question and reveals one unique challenge when intersecting register data with genotype data – the latter being less amenable to re-ascertainment.

We first emphasize a distinction between the Danish population (and associated registers), as a cohort, and the iPSYCH sample used herein. The iPSYCH data is sampled from the Danish population according to a case-cohort design but was “frozen” at December 31, 2012. DNA was obtained only for these individuals initially enrolled. The DNA data cannot be readily re-ascertained and updated with the same ease as the associated register data. The iPSYCH case-cohort, then, included a case sample of all individuals diagnosed with ADHD, ASD, ANO, MDD, BP, and SCZ as of the enrollment date (2012, $N_{\text{all cases}}=57,377$), and a random population cohort sample of $N=30,000$ controls taken from a birth cohort (May 1, 1981 to 2005, $N=1,472,762$).

For this study, we used as a discovery this initial iPSYCH enrollment of schizophrenia patients (F20.0-F20.9, $N=5432$), leveraging its principled epidemiologic design to ensure valid representation of the population of schizophrenia patients in Denmark. Because enrollment ended in 2012, enough time has passed before the initiation of this work such that many more individuals have been diagnosed with schizophrenia. However, we were unable to re-ascertain DNA on these individuals, so only the subset of the Danish population that had already been enrolled in iPSYCH were available for this study. These new schizophrenia cases were thus selected almost exclusively (>90%, see table below) from the other psychiatric cases that had been enrolled in iPSYCH in 2012 (i.e., the individuals diagnosed with SCZ between 2012 and 2016 that we could study were selected for having received a diagnosis of ADHD, ASD, ANO, MDD, or BP prior to 2012).

We reasoned that splitting the merged case group randomly might actually uncover *less robust* replications because any confounding ascertainment bias in the 2016 “selected cases” would be mirrored in both the random-subsetted discovery and replication cohorts, albeit to a proportional extent.

We now support this claim with a simulation experiment described in Supplementary Section 8, demonstrating that our “gate-keeping” design appropriately controls type-I error if the replication cohort had confounding ascertainment bias, whereas the “random splitting” design may not.

Text changes:

Please see Supplementary Section 8: “Note on choice of replication sample” and the new Table S12.

“Table S12: Number of individuals with schizophrenia in the study and (expected) number of individuals with schizophrenia in the full population cohort

	follow-up to Dec 31, 2012	follow-up to Dec 31, 2016
N_{scz} in study sample	5432	5432+870= 6302
N_{scz} in full population cohort (N=1 472 762)	5432	5432+ 79*1 472 762/30 000 ≈ 9300

Here 79 is the number of individuals in the 30 000 random population sample assigned a schizophrenia diagnosis between December 31, 2012 and December 31, 2016. ”

Question from the reviewer:

6) Also why the definition/reference to supplementary material of diagnoses used is given here at the replication sample description – should it be already in the description of the primary sample?

Authors’ response:

We agree with the reviewer. This paragraph has now been moved up to the ‘data sources’ section.

Question from the reviewer:

7) When the switch from ICD-8 to ICD-10 happened? You mention that ICD-8 was use before 1994 and ICD-10 after 1994, so does that mean that both were used during 1994?

Authors’ response:

We agree with the reviewer that this was not clear in the manuscript. ICD-10 has been in effect since January 1, 1994 (Mors 2011, ref 51). This has been made clear in the manuscript. In the Supplementary Section 7 we discuss the possible impact of the change from ICD-8 to ICD-10, and we agree with the reviewer that whereas the official change was January 1, 1994, it may be worth considering that there was a transition period before the new classification system was fully implemented in clinical practice. We have expanded the discussion about the impact of cohort effects impact on our results in the supplementary note (See answer to question 9).

Text changes:

Methods

“For every individual with a schizophrenia diagnosis, we obtained the date of first diagnosis for a broad selection of diagnoses of ICD-10-F chapter 1 or 3–9 (assigned after Jan 1, 1994) or a corresponding⁶³ ICD-8

diagnosis (assigned before Jan 1, 1994; Table S1; ICD-9 was not implemented in Denmark). F43.2–48—the etiologically-defined stress disorders — were omitted.”

Question from the reviewer:

8) Do you think it is worth of mentioning that change of disease classification did not change the definition of schizophrenia?

Authors' response:

Yes, we agree this is worth mentioning, although prior studies have found that the overlap between ICD-8/9 and ICD-10 schizophrenia diagnoses can actually be modest (Jansson, 2002). However, for our study all patients received an ICD-10 diagnosis for schizophrenia and since all patients were born after May 1, 1981 the oldest individual(s) were only 13 years old when ICD-10 was implemented. Including ICD-8 codes (295.x9 excluding 295.79) in our definition of schizophrenia was therefore used to define only parental history.

Text changes:

We have now made this distinction clearer in the Methods section:

“We included all individuals with an inpatient or outpatient hospital contact discharge code corresponding to schizophrenia in the International Statistical Classification of Diseases and Related Health Problems (8th revision (ICD-8) before 1994 codes 295.x9 excluding 295.79; 10th revision (ICD-10) after 1994 codes F20.0-F20.9.” ...

“Family history of psychiatric disorders was obtained from the PCRR using the parental CPR obtained from the Danish Civil Registration System. Family history of schizophrenia was defined as ICD-8 (before Jan 1, 1994) codes 295.x9 excluding 295.79; ICD-10 (after Jan 1, 1994) codes F20.0-F20.9.”

We have added to the discussion the potential effect of the change in diagnostic criteria used in the parents and the individuals:

“Although we see evidence for the change in diagnostic system from ICD8 to ICD10 occurring in 1994 in the rates of specific comorbidities among patients, our specific results appear robust to this effect. However, in the interpretation of the parental history, it should be kept in mind that while the majority of diagnoses in the probands were assigned under ICD-10, many of the parental diagnoses were assigned under ICD-8, which does not completely overlap with current diagnostic criteria (Jansson, 2002). Future extension of this work should still be mindful of potential complications.”

References:

Jansson L, Handest P, Nielsen J, Sæbye D, Parnas J. Exploring boundaries of schizophrenia: a comparison of ICD-10 with other diagnostic systems in first-admitted patients. *World Psychiatry*. 2002;1(2):109-114. [PMC1489865](https://pubmed.ncbi.nlm.nih.gov/1489865/)

Question from the reviewer:

9) How about the classification of other mental disorders – is it possible that changes from diagnoses given in childhood (more with ICD-8) therefore differ from the later diagnoses classified with ICD-10? I found discussion of these issues from the supplementary material chapter 7, please refer to it in the main text.

Authors' response:

As the reviewer points out, we did consider that the change in diagnostic system could impact the patterns of comorbidity. To address this, we did several things. To the extent that this is possible, we matched ICD-8 diagnoses to the best corresponding ICD-10 diagnoses (Table S1). To understand the changing patterns of diagnoses in different birth cohorts, we computed the cumulative incidence of each category of diagnoses in different birth cohorts. We show these in supplementary Figure S11. In all analyses of association with

risk factors and outcomes, we included age as a covariate. In our sensitivity analysis we computed the MDS and associations with risk factors and outcomes only for individuals born before 1991 and found that most associations still persisted. This indicates that our findings were not driven by cohort effects. We have expanded the discussion of the sensitivity analysis in Supplementary Section 6 and Supplementary Section 7.5 and now refer more explicitly to these points in the results section.

Text changes:

Results:

“Further, the change from ICD-8 to ICD-10 on Jan 1, 1994, the addition of outpatient contacts in 1995 and changes in diagnostic practice may change over time more generally which could result in birth cohort effects. To address this, we calculated the cumulative incidence of each comorbidity stratified by birth year. We observed the cumulative incidence of comorbid mood disorders, autism spectrum disorders and childhood disorders increase with birth year while comorbid personality disorders decrease (For a more detailed discussion of possible cohort effects see Supplementary Material Section 7, Supplementary Figure S11). Our sensitivity analyses suggest, however, that this cannot account for our results (Supplementary Material Section 6-7).”

Supplementary Section 7.5

“In our sensitivity analyses (Section 6, Table S11), we selected only patients born before January 1, 1991 (i.e. with more than 25 years follow-up). We looked only at their trajectory up to age 25. In Table S11 we show the results of these analyses. The 'with imputation' column shows the results using the same dissimilarity estimates as in the main analysis but doing the multidimensional scaling and association with risk factors and outcomes analyses in the >25 year follow-up subsample only. Since all individuals in this subsample were born before 1991, these analyses should be less susceptible to the possible cohort effects discussed above. Here we see that while most associations remain, the education PGS is not found, possibly related to the fact that we have relatively few samples with genotypes available in this older part of the cohort (N=1024).”

Question from the reviewer:

10) Were psychiatric outpatient and inpatient data (both) completely available since the 1981? Actually, there was an answer to that in the next paragraph and outpatient visits were available since 1995. Do you think that there is a difference/selection in your patient population due the fact that only psychiatric inpatient visits were available until 1995? There was some discussion in the supplementary material chapter 7, please refer to it.

Authors' response:

As the reviewer points out the inclusion of out-patient contacts in the Psychiatric Central Research Register started on Jan 1, 1995. We agree with the reviewer that this was unclear in the submitted manuscript and have now made it clearer.

Regarding the selection of individuals for the study, this will have very limited impact since they were selected based on having a schizophrenia diagnoses since all individuals were born after May 1, 1981, only few even the oldest patients would not have been practically eligible for a schizophrenia diagnosis before 1995 (born 1981, so 14 years old in 1995). However, it may have impact of the record of other diagnoses in these patients such as depression, childhood disorders, and others diagnoses where the severity or age of onset is more variable. We agree this could introduce some cohort effects and can see this potential in Figure S12, but note the sensitivity analyses presented in Table S11 suggest our results are robust. We now more directly reference these issues and our analyses in the main text.

Text changes:

See changes in answer to question 9.

Question from the reviewer:

11) In the linkage of family histories, which years were included? Were there a lot of missing links to fathers in CPR? Do you think that maternal or paternal mental disorders were completely captured due to the restrictions of PCRR data availability?

Authors' response:

We agree with the reviewer that the links to parents and the left-truncation the parental phenotypes were not clear in the submitted manuscript and have now added more details. The parental phenotypes were covered from 1969.

Of the 5,432 individuals, only 86 had unknown paternal history, and 80 had unknown maternal history. In total, of the 605 186 parental person years, 111 264 occurred before the beginning of PCRR on Jan 1, 1969, but only 30 800 (5%) occurred in individuals over age 10. Overall, we feel we had sufficient coverage of parental phenotypes. These numbers have now been added to notes under supplementary Table S7.

This comment uncovered a mistake in our labeling of supplemental table S7, which we have now corrected.

Text changes:

Table S7:

^e Parents were identified through the Civil Registration System and parental diagnoses obtained from the National Patient Register. Information on maternal infections during pregnancy was defined as a maternal diagnosis of infection nine month prior to the date of birth that corresponded to the gestational age of the child. Unknown parental history reflects individuals for which the father was unknown (N<80), the parent had died or left the country without a preceding diagnosis (N<10), or the parent id had been changed between the 2012 and 2016 data freezes (N<10). Of the 605 186 parental person years, only 111 264 occurred before Jan 1, 1969, where the recordings from PCRR starts (Mors, 2011), and of those only 30 800 person years (5%) occurred in individuals over age 10.

Correction of the labels in table S7:

Paternal Diagnosis, Schizophrenia ^e	no	5232
	yes	114 86
	unknown	86 114
Paternal Diagnosis, Any Psychiatric ^e	no	4251
	yes	1095
	unknown	86
Maternal Diagnosis, Schizophrenia ^e	no	5210
	yes	142 80
	unknown	80 142
Maternal Diagnosis, Any Psychiatric ^e	no	3980
	yes	1372
	unknown	80

Question from the reviewer:

12) In the survival analysis paragraph, could you please clarify/confirm that follow-up was from birth until the diagnosis of schizophrenia. So no censoring due to deaths, migration or end of follow-up were present?!

Why to use survival analysis in such a case? Or do you mean that you had eight models, i.e. one for each comorbidity category and follow-up was from birth to that particular comorbidity occurrence (or censoring due to death, migration or the end of follow-up) and if that comorbidity was not the first one, the first one was used as a time-dependent covariate? If there is more thorough description of the analysis in supplementary material, please refer to it here. I have no doubts that the analysis would be inadequate, but the description is so brief that it is difficult to follow what was actually done.

Authors' response:

We agree with the reviewer that the description of the definition of censoring in the survival analysis and cumulative incidence (question 19) was inadequate in the submitted manuscript. We considered an individual censored at whichever came first, emigration (N=65), death (N=154) according to the civil registration system, or end of follow-up (N=5213). No other competing risks were considered in these analyses (see answer to question 21). We have added this to the description of the multistate survival analysis in Methods

Text changes:

Methods:

"Cumulative incidence and Survival Analysis

Cumulative incidence was computed separately for each of the eight comorbid disorder categories. We considered an individual censored on the date of whichever came first of emigration (N=65) or death (N=154) according to the civil registration system or at the end of follow-up (Dec 31, 2016).

Within the primary cohort diagnosed with schizophrenia before December 31, 2012 we computed the hazard ratio of getting a diagnosis from each of the eight comorbidity categories given each of the other seven categories, using a Cox Proportional Hazard model, defining censoring as described above, with the first diagnosis as a time-dependent covariate and adjusting for age and sex. Apart from the two diagnoses being modelled, other diagnoses were not included in the model."

Question from the reviewer:

13) In the description of sequence analysis, there appears to be some inconsistencies in relation to supplementary material: Observation lengths: 16-36 years vs. 16-35 years and number of states (247 vs 193). Please clarify.

Authors' response:

Sequence lengths: Since the cohort consisted of individuals born after May 1, 1981 and was followed until Dec 31, 2016, the maximum follow up is 35.75 years. The standard approach in TraMineR software is to construct the sequences only for the years with complete follow up (i.e maximum length will be 35 years), however, when imputing the censored data, we felt we could leverage the information recorded between a person's last birthday and December 31, 2016. (If they received additional diagnoses in this period, it could have substantial impact on the imputation probabilities). We therefore chose to impute all sequences up to length 36.

Number of states: Since we have eight categories there are $2^8=256$ possible combinations. (no diagnosis + eight single diagnosis + 247 multiple diagnoses). Of these 256 possible states only 193 states were observed.

These distinctions have now been made clearer in the Methods and in the Supplementary Materials.

Text changes:

Methods:

“As the sequences had varying observation lengths (16–35 years) depending on subject year of birth, for all sequences we computed the probabilities of later states up to length 36 using a first order Markov chain based on population-wide estimates of transition probabilities (See Supplementary Material Section 2-3)”

In Supplement Section 2:

“Using the first event of a diagnosis with one of the eight categories of psychiatric diagnoses outlined in Table S1 and increments of 12 months, comorbidity trajectories were transformed into sequences. In case of multiple comorbidities, alphabet extensions were used (For illustration, see Figure S1). Among the 5432 included subjects, the raw (unimputed) sequences have lengths ranging from 16 to 35 years. The alphabet includes 193 unique, observed states (out of the 256 possible unique states).”

In Supplement Section 3.4:

“When imputing the sequences, the last state before censoring was set to the accumulated set of diagnoses at censoring (i.e. if one or more diagnoses had occurred between the last birthday and the end of follow-up, state used for imputation was different than the last state in the observed (unimputed) sequence). To preserve this information, also in the longest sequences, all sequences were imputed up to length 36 years.”

Question from the reviewer:

14) Please describe how state for each year was defined, i.e. was it required that comorbid diagnoses occurred during the same year (i.e. person had to have admissions with those diagnoses at the considered year) or were diagnoses cumulative so that after the first diagnosis that diagnosis remained for all subsequent years?

Authors' response:

We defined the states as the complete set of diagnoses an individual had received before the beginning of that year (of the person's life). This has now been made explicit in the methods section.

Text changes:

Methods:

“Each patient's psychiatric medical history was transformed into a sequence of states, wherein each state corresponded to a period of one year, from birth to the end of 2016 (Figure 3). The sequence alphabet consisted of a no diagnosis-state, a state for selected diagnoses from each of the eight diagnostic chapters (Table S1), and a state for each of the 247 possible combinations of diagnoses. For each year until the end of follow-up, an individual's state was defined as the cumulative set of diagnoses they had received before the beginning of that year.”

Question from the reviewer:

15) The technique for imputing the missing states is well described and providing the software to do that is a definitively positive thing. However, it seems that most sequences needed imputation and partly more than half of the whole sequence was imputed. In general, the predictions to near-future tend to be more accurate than the ones for the distant future. What do you think, is the case similar here, i.e. that the

accuracy of the imputation is getting “worse” for longer imputation periods? Is there anything you could do about it?

Authors' response:

This is a good point. We did not mask observed trajectories in this data, however, prior to the analyses, we used an independent, general purpose reference data set to build our imputation protocol (data and code not shown here, but now available at:

<https://github.com/MortenKrebs/diagtraject/blob/master/README.md>). These analyses confirm the reviewer's intuition in that imputation accuracy decreases the further away in time we get from the end of follow-up.

When computing the dissimilarities, we use the weighted mean dissimilarity of all the possible states an individual can be in, given the last state before censoring. This will lead to a shrink in the dissimilarities for those individuals with the shortest follow-up. We cannot exclude that other imputation methods than the Markov Chain approach undertaken here can improve the quality of the imputation, but the general tendency of the accuracy to get lower the further you go from the end of follow-up will likely remain. However, when testing for association with outcomes, a reduction in the variance of the dissimilarities should only bias us towards the null.

To address this, one could partially censor our real, complete trajectories and assess the imputation here in, but given the sensitivity analyses that were restricted only to complete trajectories (supplement chapter 6, now expanded; please see below) gave concordant results to those with the imputed trajectories, we did not feel this was necessary.

Question from the reviewer:

16) You describe some sensitivity analyses in supplementary material chapter 6, but no results of these analyses or reference to them is presented there?! There appears to be some results in supplementary table 11

Authors' response:

We agree that the description of these sensitivity analyses was very brief and have now added a more detailed presentation of the results of these analyses in the Supplementary Section 6.

Text changes:

Supplementary Section 6:

“In Table S11 we show the results of these analyses. The 'with imputation' column shows the results using the same dissimilarity estimates as in the main analysis but doing the multidimensional scaling and association analysis in the >25 years follow-up subsample only. Here we see that while most associations remain, the education PGS is not found, possibly related to the fact that we have relatively few samples with genotypes available in this older part of the cohort (N=1024).

Next, in the 'OM_jacc_1' column, we use the same alignment method, substitution cost and indel cost, but look only at the part of the trajectories that are fully observed (i.e., we compare sequences with lengths of exactly 25 years). Here we see that most associations attenuate a little but that the overall pattern seems unchanged, which could be ascribed to the loss of information from censoring all individuals at age 25.

Varying the indel costs have very small effects, indicating that the alignment is driven largely by substitutions.

Defining the substitution costs as the simple matching coefficient rather than the Jaccard distance (which is equivalent to multi-channel sequence analysis with two states per channel) also doesn't change the general picture of associations, but a constant substitution cost (i.e. ignoring that some states consist of

overlapping combinations of diagnoses and treating all as equally similar) seems to have a bigger impact, but also gives similar patterns of association and here, indels have the potential to influence the alignment more, but also does not seem to change the overall pattern much.

The Euclidian distance between the probability distributions is an alternative measure, but when the number of distributions is high it approximates the hamming distance-based alignment procedures with a constant cost (Studer 2016). In accordance with this, we also found the same general patterns of association. The chi2 metric puts weight on the rarity of a state when the dissimilarities are computed and have previously been shown to be fundamentally different from all other sequence analysis methods applied here (Studer 2016), none of our associations were seen when the chi2 based metrics were used, probably since the associations we find were associated to differences in the sequences of common states primarily. Taken together, we find that the reported associations are robust across a range of alignment methods and parameter choices."

Question from the reviewer:

Nice analyses for the stability of the MDS.

17) In the description of MANCOVA analyses, please specify how many dimensions of MDS were used (all 13 or the first 3 or something else)?

Authors' response:

We used the three MDS dimensions as dependent variables in the MANCOVAs. This is now also made explicit in the methods section.

Text changes:

Methods:

"Multivariate analysis of covariance (MANCOVA) was used to test for association between selected MDS dimension scores of dimensions 1-3 (dependent variables) and 18 putative risk variables from the registries, including pregnancy and birth-related variables, hospital contacts due to an infection both in the individual and maternal infections during the pregnancy period, and family history of psychiatric disorders (Table S7). MANCOVAs included covariates for age at the end of the study period and sex." ...

"These five PGSs and the count of rare mutations were used in MANCOVAs with MDS dimensions (1-3) as dependent variables while adjusting for the first ten principal components (PCs) of genetic ancestry, genotyping wave, age, and sex."

Question from the reviewer:

18) In the multinomial logistic regression, how many clusters were used (was it five)?

Authors' response:

Five clusters were used for the multinomial logistic regression. This is now also made more explicit in the methods section.

Text changes:

Methods:

"A multinomial logistic regression was conducted to test for associations between the variables with significant association in the MANCOVA and cluster membership in the selected (k=5) clustering."

Question from the reviewer: |

19) In "association with outcome" paragraph, were your dependent variables ("the disease trajectories") the MDS dimensions (and how many) again? If so, please write it out.

Authors' response:

This is now also made explicit in the methods section.

Text changes:

Methods

"Additionally, we tested if disease trajectories described by the selected MDS dimensions (1,2, and 3) were associated with the average yearly number of hospital admissions under a schizophrenia diagnosis after the first diagnosis of schizophrenia and the average number of days spent in hospital per year. For these analyses we used MANCOVAs as described above adjusting for age and sex.

Question from the reviewer:

DATA AND CODE AVAILABILITY

Excellent shiny app for further results and detailed R code for most of the analyses provided.

RESULTS

20) Here it is stated that 10864 population controls (i.e. two per cohort member) were also followed. Why this has not been mentioned in the methods, abstract or supplementary material?

Why there are (minor) differences in the age and sex distributions even though the controls were matched? And please clarify: matched at which point in time (probably at the time of schizophrenia)?

Authors' response:

We agree with the reviewer that the procedure for matching cases to population controls was not well detailed in the manuscript. We have added the description to the Supplementary Material and refer to it in the Methods in the main text. As to the minor differences between the cohorts, exact matching was not possible, so we used propensity score matching which allows for some imperfections in the matching of the age and gender distribution. Details about the matching are now outlined in Supplementary Section 1.1 and we refer to it in the methods section.

Text changes:

Methods:

"For every individual with a schizophrenia diagnosis and 10 864 age and sex matched population controls (Supplementary Section 1.1) we obtained the date of first diagnosis for a broad selection of diagnoses of ICD-10-F chapter 1 or 3–9 (assigned after Jan 1, 1994) or a corresponding⁶³ ICD-8 diagnosis (assigned before Jan 1, 1994; Table S1; ICD-9 was not implemented in Denmark). F43.2–48—the etiologically-defined stress disorders — were omitted."

Supplementary Section 1.1:

"Matched controls

To identify population controls with similar characteristic to the case sample, we sampled from the non-overlapping part of the (N=30 000) iPSYCH population sample, matching on age and sex. Matching was done using propensity score matching as implemented in the r function 'MatchIt::matchit' (Ho, 2011) with matching on age at end of follow up and sex (settings 'method="nearest"', 'distance="glm"' and 'ratio=2'). We observed a standardized mean difference (Austin, 2009) in the covariates of 6% for gender and 0.7% for age, which we considered acceptable as they were below the typical 10% recommendation (Austin, 2009). As outlined in Table 1 this resulted in a (N=10 864) control cohort with similar age and gender characteristic to the case sample."

References:

Ho DE, Imai K, King G, Stuart EA (2011). "MatchIt: Nonparametric Preprocessing for Parametric Causal Inference." Journal of Statistical Software, 42(8), 1–28. <https://www.jstatsoft.org/v42/i08/>

Austin PC. Balance diagnostics for comparing the distribution of baseline covariates between treatment groups in propensity-score matched samples. *Stat Med.* 2009;28(25):3083-3107. doi:10.1002/sim.3697

Question from the reviewer:

21) In the figure 1, cumulative incidences are obviously calculated separately for each disorder as summing those up (to “combined outcomes”) gives proportions over 1. Was potential censoring (due to death, migration or end of follow-up) after diagnosis of schizophrenia taken into account? Recently it has been claimed that cumulative incidence should be calculated by taking the competing risks into account to provide reasonable estimates – how would you explain why that approach has not been used here (at least for the part after schizophrenia/matching date)?

Authors' response:

We apologize for these inadequate descriptions and have added substantial details to the Methods sections (See answer to question 12)

Question from the reviewer:

22) Typo in line 125: $p < 0.000$ -> $p < 0.0001$

We had addressed this error in the revised manuscript. We used a Bonferroni correction for 56 tests.

Text changes:

$p < 0.00089$

Question from the reviewer:

23) Secondary comorbidity analyses in Figure 2. Description of the Cox models was somewhat unclear (see comments in the methods section) and also the R-code for these analyses is a bit difficult to read without testing it with some data. As mstate-package was used, were you taking competing risks into account in these analyses? Please add a bit more specific description to the methods section.

Authors' response:

We agree that this was inadequately described. We have now expanded the description of the Cox models (answer to in question 12) to better describe the decisions about censoring. While we agree that modelling the hazards of the diagnoses two at a time is unsatisfying since other diagnoses might work as competing risks, we feel that the multistate survival analysis is best suited for a limited number of states and that modeling all the possible transitions between all (256) states resulting from combinations of eight diagnoses will be to complex a model to fit in a survival framework. This was why we chose the sequence analyses undertaken in the subsequent parts of the paper. If there are particular states/diagnoses that the reviewer thinks it would be worth including as competing risks in targeted sensitivity analyses, we would be happy to provide those results.

Question from the reviewer:

24) Line 141: You refer to Figure 1 while describing psychiatric trajectories. Do you mean Figure 3?

Yes, we have addressed this error in the revised manuscript.

Question from the reviewer:

25) What are the “Online Methods” representing here – supplementary material or R-code? If supplementary material, please refer to it so that the corresponding section can be identified from the material.

Authors’ response:

We were following the naming conventions of Nature Communications articles which place the general Methods section in the online version, only, and refer to them as “Online Methods.” We ensured this was consistent throughout and distinguished from reference to supplementary text, tables, and figures (Supplementary Material, Supplementary Figures and Supplementary Tables) and our online code (url).

Question from the reviewer:

26) Please specify what correlations represent while you lower time increments as reported in lines 147-148. I understand that you have dissimilarity-matrix between trajectories, but hasn’t that matrix different dimensions if you change time increments so how to calculate correlation?

Authors’ response:

We agree with the reviewer that the descriptions of these analyses were very brief. The dissimilarity matrix always has the same dimensions, because we compute it person vs. person (5432x5432), regardless of the lengths of the considered trajectories. What we report is the Pearson correlation between all the off-diagonal entries of the matrix computed with one-year increments and those computed with smaller increments. This means that individuals are comparatively alike or not alike, when considering trajectories of different intervals.

Text changes:

This has now been made more explicit in the Results section:

“To test if lowering time increment size would affect the results, we computed the sequence dissimilarities using six and four months increment and computed the correlation of the lower triangular entries of the resulting dissimilarity matrices with those obtained using one-year increments. We found the correlations to be high ($r_{\text{Pearson}}=0.9991$, $r_{\text{Pearson}}=0.9987$, respectively) and therefore proceeded with the one-year increment size.”

Question from the reviewer:

27) In the results section (line 150) it is stated that logistic regression analyses were conducted, which sounds clear for diagnoses, but in the analyses for total count of diagnoses or the age at first diagnosis you probably have utilized other than logistic regression?

Authors’ response:

The reviewer is right that this reflects a typo. As shown in Supplementary Figure S5, we did use linear regression for the continuous and pseudo-continuous variables (counts of disorders). This was chosen after a comparison of fit with a more conceptually aligned Poisson regression (Supplementary Figure S5).

Text changes:

“...To assess individual dimension loadings, we performed post hoc linear and logistic regression analyses, predicting, in series, the constituent diagnoses (logistic regression), their total count (linear regression), and age at first diagnosis (linear regression), from each of the three principal MDS dimensions, independently.”

Question from the reviewer:

28) The paragraph describing analyses for post hoc analyses giving interpretations to MDS dimensions is somewhat complicated to read as different approaches or scales have been reported for different

dimensions. Those seem to be fine, but the question is how those interpretations were found and selected to be the best ones?

Authors' response:

We apologize for this confusion and agree there is a logical step missing. Our paper is quite burdensome in terms of figures and analyses, so we omitted a less technical intermediate step. We have now re-introduced an original figure as supplemental Figure S4. This figure is a descriptive plot that shows visually the distribution of features across MDS components and visual inspection was used to select a set of hypotheses for more formal testing.

We now mention this intermediate step in the results section.

Text changes:

In the Results:

"To assess individual dimension loadings, we visualized the patterns of comorbidity at different quantiles of each dimension (Figure S4) and, based on these patterns, we performed post hoc linear and logistic regression analyses, predicting, in series, the constituent diagnoses (logistic regression), their total count (linear regression), and age at first diagnosis (linear regression), from each of the three principal MDS dimensions, independently.

In the Supplement we added this figure as Figure S4

With figure legend:

"The first row displays the prevalence of different number of different categories of diagnoses (other than schizophrenia) as defined in Table S1 in each 2-percent quantile of each of dimensions 1-7. Rows 2-9 display the frequency of each of the categories of psychiatric diagnoses with different age on onset in each 2-percent quantile of each of dimensions 1-7. Row 9 displays the frequency of different onset ages of schizophrenia for each 2-percent quantile of each of dimensions 1-7."

Question from the reviewer:

29) In Figure 4B, what does darker grey in cluster 1 represent (missing from the legend)?

Authors' response:

Grey color indicates that at this time-point the most common state was the 'censored' state. We have now updated the figure so there are no longer two shades of grey and added the censored state to the legend.

Question from the reviewer:

30) As the clustering results in relevantly stable and interpretative groups of patients, would it be possible to give certain fixed definitions for these groups so that the similar division to groups could be done based on those rules instead of using data-driven clustering? That would allow much better repeatability of the analysis in other contexts just by assuming that groups are somehow reasonable from clinical point of view while repeating whole data-driven approach will certainly result in different groupings masking the possibility for direct comparisons.

Authors' response:

We agree with the reviewer that it is a limitation that it is not possible to classify external cohorts according to our clustering without access to our data, but due to the sensitive nature of our data we cannot make data publicly available at the person-level. However, we agree that enabling external replication is a critical part of quality science and have derived what appears to be a robust solution.

We have now added a figure (Supplementary Figure S14) where we developed a “best fitting” decision tree that aims to approximate our MDS-clustering classification of patients directly from their observed diagnoses and timing data. The tree-based grouping captures a large portion of the information from our more detailed approach, with the resulting groupings showing high overlap with the MDS-based clusters (rand index 0.83). Importantly, associations with risk factors and outcomes are highly concordant between the two clustering approaches, suggesting this could be a viable approach for external groups to pursue independent replications.

A description of the tree-generation, the resulting decision tree, and the concordance of risk factor and outcome associations has been added to the supplementary material.

Text changes:

We also added these analyses to the Results section:

“In order to enable external replication and use of these clusters, we trained a decision tree on the k=5 MDS-based clustering that produces a largely overlapping classification (rand index=0.83; Supplementary Figure S8) with highly concordant associations to risk factors and outcomes (Supplementary Figure S9).”

Supplementary Figure S14 with the legend:

“A decision tree was trained to classify patients into the groups described by the k=5 clustering based on the presence/absence of each of the eight categories of diagnoses (Table S1) and on their age at onset. This was done using the rpart software (<https://cran.r-project.org/web/packages/rpart/>) treating the clustering as categorical outcome (method='class'). The resulting tree presented above”

Supplementary Figure S15 with the legend:

“Multinomial logistic regression is conducted using either MDS-based cluster membership (x-axis) or decision tree classification (y-axis) as dependent variable sequentially using the predictor variables as independent variable and treating Cluster 5 as reference. All regressions are adjusted for age and sex. PGS - Education is adjusted additionally for 10 principal components and genotype wave. Error bars indicate 95% confidence intervals.”

Question from the reviewer:

31) Is there a need to report all F and P values for MANCOVA and ANCOVA analyses if those are in any case reported in Table 2.

Authors' response:

We have removed the F values from the text and instead refer to Table 2.

Question from the reviewer:

32) In addition, I would not in this case use scientific notation for P-values in text or (supplemental) tables.

Authors' response:

We assume the reviewer is referring to Results and Table 2 where we use the “5e-02” type of notation. We have now changed the notation in this table.

Question from the reviewer:

33) I would also prefer reporting confidence intervals at least with OR:s and HR:s.

Authors' response:

We have expanded Table S2 to include the HR, 95% confidence intervals and p-values from Figure 2. In Table S8 we have added 95% confidence intervals to the odds ratios.

Question from the reviewer:

34) Sensitivity analyses are comprehensive. Would it be possible to somehow indicate potential inconsistencies in the supplementary table 11? Now it is difficult to read.

Authors' response:

See answer to question 16. We now added a paragraph in Supplementary Section 6 summarizing the results of the sensitivity analyses in more detail.

Question from the reviewer:

DISCUSSION

I think the discussion is comprehensive and well structured. Just a few ideas that emerged while reading it:

35) Are there some psychiatric comorbidities that are particularly strongly associated with later schizophrenia? For example, it has been reported that half of patients with cannabis-induced psychosis have got schizophrenia diagnosis in five years.

Authors' response:

This is a very interesting question. An analysis of it would involve bringing additional patient cohorts into this study (for example, looking at all individuals with childhood disorders and comparing the conversion rates). While interesting, it is a bit beyond the scope of this work, these analyses were conducted by our colleagues (see Plana-Ripoll et al. (2019), doi:10.1001/jamapsychiatry.2018.3658, in particular eFigure 3 right panel) where they showed that while a diagnosis from any ICD-10-F chapter increases the risk of a subsequent schizophrenia diagnosis, the relative risk is particularly high after a substance abuse diagnosis.

Question from the reviewer:

36) Would it possible to include birth season as a risk factor for schizophrenia here? It has been reported that birth rate for people with schizophrenia is higher during spring or winter months.

Authors' response:

The original finding (Mortensen et al. NEJM, 1998) was made using the schizophrenia patients diagnosed according to ICD8 criteria and internal attempts at replication with ICD-10 defined schizophrenia have suggested a less robust association (data not shown, unpublished). Therefore, we did not include season of birth in our analyses, however, it could, and perhaps should, be reconsidered in future work.

From the reviewer:

In conclusion, the study appears to be very elegantly conducted with refreshingly different methods with novel results. There are some shortcomings in the reporting (how the manuscript has been written) and I have tried to point out issues that complicated my reading of the manuscript. I hope that those comments/suggestions help to improve the (readability of the) manuscript further, but I left it up to the authors (and editor) to decide which ones require corrections.

Dr. Reijo Sund
Professor of Register Studies,
University of Eastern Finland,
Kuopio, Finland

Reviewer #5:

Dear Reviewer 5,

We thank you for your careful read of our work, in particular the connections to the work of Dickinson et al which provides a nice external concordance for our data-driven results. Note, changes from the pre-existing text are highlighted in yellow below.

Question from the reviewer:

I gather that this manuscript has been through at least one round of reviews and revisions and I will keep this in mind. However, this is my first review of the paper.

Authors present findings from a remarkable, longitudinal sample of >5400 Danish schizophrenia cases based on detailed information from Danish civil and health registries. They characterized these individuals in terms of sequences of comorbid diagnoses arising before, after, and concurrently with the focal schizophrenia diagnosis, thus creating developmental comorbidity trajectories for each case. The comorbidity trajectory data were then analyzed using MDS to identify prominent comorbidity trajectory dimensions and, further, were subjected to potentially simplifying cluster analyses to identify trajectory subgroups. Finally, authors tested the associations of both MDS dimensions and cluster-based subgroups with genetics (PGS), gestational factors, parental factors including mental health history, and certain indicators of clinical severity and course. Results highlighted the importance of (1) the number of comorbid diagnoses, (2) diagnoses of childhood disorder comorbidities vs mainly adult comorbidities, and (3) the presence of a substance abuse comorbidity. Clinical outcomes, gestational infections and other gestational issues, family history of mental illness, and education PGS, among other things, were differentially associated with dimensional differences and subgroup membership.

Heterogeneity is a challenge for clinical care and research in schizophrenia with huge implications for biological and treatment discovery. In particular, heterogeneity in developmental and clinical course in this disorder is widely recognized and fundamental. The current analyses take a novel approach to this aspect of schizophrenia heterogeneity and offer possible clues to risk, etiology and clinical outcome. To me the analytical approaches are well-matched to the data and research goals. I particularly appreciated seeing associations – for both dimensional and categorical approaches – to the family history, gestational, hospitalization, and genetic variables. My principal comments relate to discussion points.

1) The paragraph beginning on line 351 discusses the association of education PGS with the dimensional and categorical structure of comorbidity trajectories. One recent study (Dickinson et al., American Journal of Psychiatry, 2020) reported consistent findings that authors might wish to reference. Dickinson et al. found that a profile of PGS, including educational PGS, distinguished schizophrenia subgroups defined in terms of putative trajectories of cognitive development. In particular, education PGS helped distinguish a subgroup characterized by suspected early childhood impairment from other groups with different developmental trajectories, consistent with certain findings here.

Authors' response:

We thank the reviewer for pointing us to this interesting study by Dickinson et al. We agree there are several findings in the study that are consistent with their findings. We now cite this paper in the discussion and discuss these similarities:

Text changes:

"This work is also relevant to the position of schizophrenia as a neurodevelopmental disorder.³⁵ The second dimension, which was in particular capturing patient trajectories denoted by increased prevalence

of comorbid childhood psychiatric diagnoses could reflect predominance of an earlier developmental pathway that presents in childhood, possibly overlapping with what has been described as premorbid schizophrenia symptoms³⁵. This is in line with a recent study by Dickinson et al. (Dickinson, 2020) who used a cross-sectional design and found a subgroup of schizophrenia patients with signs of a more neurodevelopmental course. This dimension was associated with a maternal schizophrenia diagnosis. Notably, the pioneering work by Fish et al.³⁶ described the developmental *pandysmaturation* syndrome among newborns of mothers with schizophrenia. However, this association could also have other explanations, e.g. mother's diagnosed with schizophrenia have more frequent contact with hospitals and may receive additional monitoring from the healthcare sector such that childhood symptoms receive early medical attention. This dimension was somewhat surprisingly associated with a less severe disease course after the schizophrenia diagnosis as measured by number of admissions. Interestingly, Dickinson et al. (Dickinson, 2020) also found that individuals with schizophrenia and a pre-adolescent cognitive impairment had less severe symptoms after onset, than those with more typical pre-adolescent cognitive performance that declined (adolescent disruption of cognitive development). Also, this could be due to the fact that this dimension was associated with less exposure to other known risk factors. Both dimension 1 and 2 could co-adhere with subtypes of schizophrenia with a more neurodevelopmental components, where particularly dimension 1 has a higher levels of early life risk factors for schizophrenia, which could indicate that neurodevelopmental pathology plays a particularly important role for these dimensions."

Question from the reviewer:

2) Authors have marshalled and aligned large streams of longitudinal data to conduct these very interesting analyses. Limited information about outcomes, however, constrains interpretation. Authors are able to make interesting connections to hospitalization variables. Information about functional outcomes, including educational and/or vocational attainment by schizophrenia cases, would have been a valuable complement.

Authors' response:

This is a really interesting suggestion by the reviewer, and we agree that many of our "risk factor and outcome" associations emphasize potential causes of heterogeneity within schizophrenia, which was our conceptual focus here. We agree that it would be both highly relevant and informative to pursue a deeper analysis on the consequences of this heterogeneity on longer term and broader outcomes in, for example, the labor market and intend to pursue such studies in the future. It does remain slightly beyond the scope of this work, however, because of the current young age of our cohort and additional linkage required to access these data in registers.

Text changes:

We have now emphasized this point in the Discussion:

"but we feel that more far-reaching searches, including other measure of social environment (e.g. household income or parental education or marital status), copy number and structure variants, broader collections of clinical factors, and PRS from multiple traits and diseases are needed to more fully characterize the nature of disorder course heterogeneity in schizophrenia. In addition, future work could emphasize longer term patient outcomes, such as participation in the labor market."

Question from the reviewer:

3) For example, around line 291 authors indicate surprise that the subgroup with comorbid childhood disorders showed less severe hospitalization outcomes after diagnosis with schizophrenia. However, relatively lower clinical instability in this group might be accompanied by significant limitations in social and role functioning in the community (eg, Dickinson et al., 2020 – moderate symptomatology, but lower levels

of education, employment and cognitive performance in early life impairment group). Perhaps these data were not readily available from public registry sources, but this seems a potentially fruitful direction for future work.

Authors' response:

We agree that our finding that individuals with trajectories characterized by childhood disorders appear to have fewer hospitalizations is in line with the findings by Dickinson et al. and have now included this point in the Discussion (see answer to question 1).

Smaller points:

4) Regarding line 167 'nearly half of between-patient variability in trajectory dissimilarity' – Is this accurate? Or is it nearly half of the 79% trajectory dissimilarity variance explained by the 3 MDS dimensions?

Authors' response:

We agree with the review that this was incorrect. We have changed the sentence.

Text changes:

We found that nearly half of the variance in the three MDS scores was captured by five clusters ($R^2=0.48$; Figure S7) with distinct and clinically interpretable characteristics.

Question from the reviewer:

5) Column B in Figure 4 is a bit confusing. It would be better if there were not two shades of gray in this part of the figure. I assume that the pale gray color signifies that the modal diagnosis for that particular follow-up period is schizophrenia. However, the meaning of pale gray color should be stated explicitly in the color key at the bottom of the column.

Authors' response:

We agree with the reviewer and have changed it accordingly.

Question from the reviewer:

6) More generally, I found the figure captions overly terse and that they could benefit from revisions for greater clarity.

Authors' response:

We agree and have changed the captions to be more informative.

Text changes:

"Figure 1 | Cumulative incidence at age 30 of eight comorbid psychiatric disorder categories in a Danish cohort of 5432 individuals with schizophrenia and 10864 random population controls"

"Figure 2 | Hazard Ratio of eight comorbid psychiatric disorder categories conditioned on a prior diagnosis of each of the seven other psychiatric disorder categories estimated in 5432 individuals with schizophrenia"

"Figure 3 | Schematic of the Sequence Analysis procedure used to compute the dissimilarity of psychiatric comorbidity trajectories"

"Figure 4 | Results of the multidimensional scaling and clustering based on dissimilarity of psychiatric comorbidity sequences in schizophrenia and cumulative incidence of comorbid psychiatric disorder categories and levels of risk factors and outcomes in each of the five clusters identified"

Question from the reviewer:

7) Table S 8, should there be columns for Cluster #5?

Authors' response:

We agree with the reviewer that this was unclear in the submitted manuscript. Cluster 5 was treated as reference in the multinomial logistic regression, so the odd ratios are all relative to Cluster 5. We now explain this in the methods and in the table notes under Table S8.

Text changes:

Table S8 legend:

"Multinomial logistic regression is conducted using the cluster membership as dependent variable sequentially using the predictor variables as independent variable and treating Cluster 5 as reference. All regressions are adjusted for age and sex.

PGS - Education is adjusted additionally for 10 principal components and genotype wave.

Global assoc. test = likelihood ratio (LR) test comparing full model to a model with all covariates, but without the predictor."

Methods:

"A multinomial logistic regression was conducted to test for associations between the variables with significant association in the MANCOVA and the clusters. These were conducted sequentially with the cluster membership as dependent variable and each of the predictor variables as independent variable with adjustment for sex and age and for genetic variables additional adjustment for 10 PCs and genotyping wave. Odds ratios were reported treating cluster 5 as reference. Since these were selected based on association in the MANCOVA, we conservatively applied the same significance threshold ($p < 0.05/25 = 0.002$)."

Question from the reviewer:

8) Should the year ranges on lines 94 and 106 be consistent?

Authors' response:

When individuals with schizophrenia were only included if the diagnosis was assigned after age 10, the cohort was created per Dec 31, 2012, the cohort was limited to individuals born before Dec 31, 2002, which is in contrast to other iPSYCH cohorts that include individuals born before Dec 31, 2005. We have now changed this to be consistent throughout the manuscript.

Question from the reviewer:

Line 150 'assess'

Line 240 'sensitivity'?

Line 291 '... what have been described ...'

Line 424 'PRS'? Or 'PGS', as used elsewhere.

Line 763 '... each to be assigned a cost.'?

Now rephrased:

“Alignment could be obtained by substitutions, insertions and deletions which were each assigned a specific cost”

Authors' response:

Thank you for catching these. We have addressed them in the revised manuscript.

REVIEWERS' COMMENTS

Reviewer #4 (Remarks to the Author):

The authors have adequately reacted to my earlier remarks. I have no more comments.

Dr. Reijo Sund
Professor of Register Studies,
University of Eastern Finland,
Kuopio, Finland

Reviewer #5 (Remarks to the Author):

I think this is a very well-done paper and likely to have an impact on thinking about heterogeneity and varying developmental trajectories in schizophrenia. My concerns were few and have been addressed in this revision. I think authors have also taken reasonable steps to address the concerns expressed by the other reviewers.